# THE LARGE LEARNING RATE PHASE OF DEEP LEARNING

## ABSTRACT

The choice of initial learning rate can have a profound effect on the performance of deep networks. We present empirical evidence that networks exhibit sharply distinct behaviors at small and large learning rates. In the small learning rate phase, training can be understood using the existing theory of infinitely wide neural networks. At large learning rates, we find that networks exhibit qualitatively distinct phenomena that cannot be explained by existing theory: The loss grows during the early part of training, and optimization eventually converges to a flatter minimum. Furthermore, we find that the optimal performance is often found in the large learning rate phase. To better understand this behavior we analyze the dynamics of a two-layer linear network and prove that it exhibits these different phases. We find good agreement between our analysis and the training dynamics observed in realistic deep learning settings.

## 1 INTRODUCTION

Deep learning has shown remarkable success across a variety of tasks. At the same time, our theoretical understanding of deep learning methods remains limited. In particular, the interplay between training dynamics, properties of the learned network, and generalization remains a largely open problem.

In tackling this problem, much progress has been made by studying deep neural networks whose hidden layers are wide. In the limit of infinite width, connections between stochastic gradient descent (SGD) dynamics of neural networks, compositional kernels, and linear models have been made. These connections hold when the learning rate is sufficiently small. However, a theory of the dynamics of deep networks that operate outside this regime remains largely open.

In this work, we present evidence that SGD dynamics change significantly when the learning rate is above a critical value, $\eta_{\mathrm{crit}}$, determined by the local curvature of the loss landscape at initialization. These dynamics are stable above the critical learning rate, up to a maximum learning rate $\eta_{\mathrm{max}}$. Training at these large learning rates results in different signatures than observed for learning rates $\eta < \eta_{\mathrm{crit}}$: the loss initially increases and peaks before decreasing again, and the local curvature drops significantly early in training. We typically find that the best performance is obtained when training above the critical learning rate. Empirically, we find these two learning rate regimes are robust, holding across a variety of architectural and data settings.

Figure 1 highlights our key findings. We now describe the main contributions of this work.

### 1.1 TRAINING WITH A LARGE LEARNING RATE LEADS TO A CATAPULT EFFECT

Consider a deep network defined by the network function $f(\theta, x)$, where $\theta$ are the model parameters and $x$ the input. We define the curvature $\lambda_t$ at training step $t$ to be the max eigenvalue of the Fisher Information Matrix, $F_t := \mathbb{E}_x \left[ \nabla_\theta f(\theta_t, x) \nabla_\theta f(\theta_t, x)^T \right]$ Amari et al. (2000); Karakida et al. (2018). Equivalently, $\lambda_t$ is the max eigenvalue of the Neural Tangent Kernel Jacot et al. (2018).

Figure 2 shows the results of training several deep networks with mean squared error (MSE) loss using SGD with a range of learning rates. The loss and curvature are measured at every step during

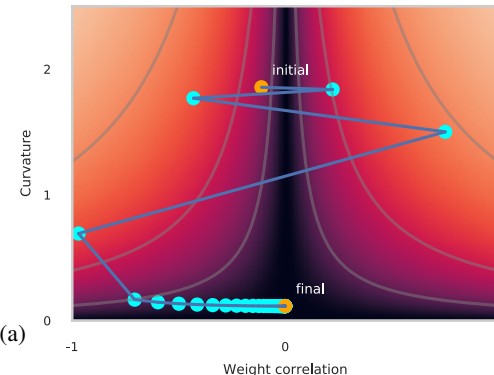
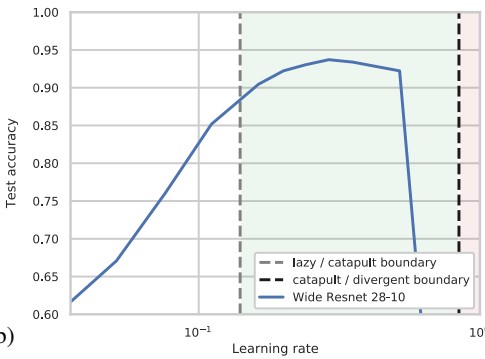

Figure 1: **Large learning rates lead to large weight movement and better performance**. (a) A visualization of gradient descent dynamics derived in our analytic model. A 2D slice of parameter space is shown, where lighter color indicates higher loss and dots represents points visited during optimization. Initially, the loss grows rapidly while local curvature decreases. Once curvature is sufficiently low, gradient descent converges to a flat minimum. We call this the *catapult effect*. See Figures S2 and S1 for more details. (b) Confirmation of our predictions in a practical deep learning setting. Line shows the test accuracy of a Wide ResNet trained on CIFAR-10 as a function of learning rate, each trained for a fixed number of steps. Dashed lines show our predictions for the boundaries of the large learning rate regime (the *catapult phase*), where we expect optimal performance to occur. Maximal performance is achieved between the dashed lines, confirming our predictions. See Section 2 for details.

the early part of training. We notice the following effects, which occur when the learning rate is above the critical value $\eta_{crit} = 2/\lambda_0$, where $\lambda_0$ is the curvature at initialization.[1]

1. During the first few steps of training, the loss grows significantly compared to its initial value before it begins decreasing. We call this the catapult effect.

2. Over the same time frame, the curvature decreases until it is below $2/\eta$.

We can build intuition for these effects using loss landscape considerations. Consider a linear model where the curvature of the loss landscape is given by $\lambda_0$. Here, curvature means the largest eigenvalue of the linear model kernel. The model can be trained using gradient descent as long as the learning rate $\eta$ obeys $\eta < 2/\lambda_0$. When $\eta > 2/\lambda_0$, the loss diverges and optimization fails.

Next, consider a deep network. If we train the model with learning rate $\eta > \eta_{crit}$, we may again expect the loss to grow initially, assuming the curvature is approximately constant in the neighborhood of the initial point in parameter space. This is the effect observed in Figure 2. However, unlike the linear case, optimization may still succeed if gradient descent is able to navigate to an area of the landscape that has lower curvature $\lambda$, such that $\eta < 2/\lambda$. This is indeed what we observe in practice.

In Figure 1 we show that optimal performance typically occurs when a network is trained in the large learning rate regime. As discussed further in Section 2, this is true even when the compute budget for smaller learning rates is increased to account for the smaller step size. This is consistent with previous observations in the literature, which showed a correlation between performance and the flatness of the minimum (Keskar et al., 2016).

## 1.2 AT LARGE WIDTH, A SHARP DISTINCTION BETWEEN LEARNING RATES REGIMES

The large width limit of deep networks has been shown to lead to simplified training dynamics that are amenable to theoretical study, as in the case of the Neural Tangent Kernel (Jacot et al., 2018). In this work we show that the distinction between small and large learning rates becomes sharply defined at large width. This can be seen in Figures 2c, 2f, which show the curvature of sufficiently wide networks after the initial part of training, as a function of learning rate. When $\eta < \eta_{crit}$ the curvature is approximately independent of the learning rate, while for $\eta > \eta_{crit}$ the curvature is lower than $2/\eta$.

---

[1]The critical learning rate depends on the scale of initialization through $\lambda_0$.

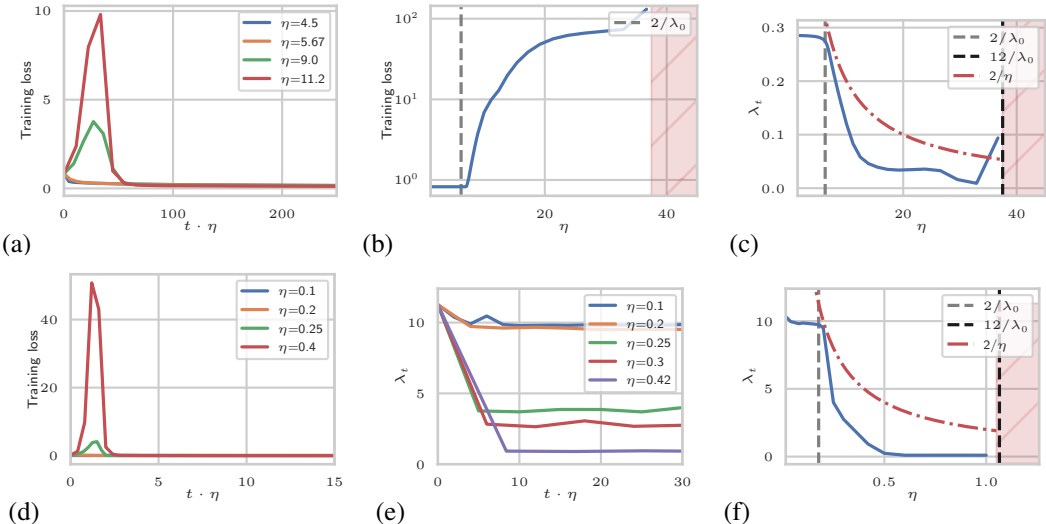

Figure 2: **Early time catapult dynamics**. (a,b,c) A 3 hidden layer fully-connected network with ReLU non-linearity with width 2048 trained on MNIST ($\eta_{\mathrm{crit}} = 6.25$). (d,e,f) Wide ResNet 28-10 trained on CIFAR-10 ($\eta_{\mathrm{crit}} = 0.18$). Both networks are trained with vanilla SGD; for more experimental details see Appendix A. (a,d) Early time dynamics of the training loss for learning rates in the linear and catapult phases. (b) Maximum value of the loss as a function of the learning rate. (e) Early time dynamics of the curvature for learning rates in the linear and catapult phase. (c,f) $\lambda_t$ measured at $t \cdot \eta = 250$ (for FC) and $t \cdot \eta = 30$ (for WRN), as a function of learning rate. Training diverges for learning rates in the shaded region.

In Section 3 we analyze the gradient descent dynamics of 2-layer linear networks, and find that they exhibit similar behavior that can be understood in the large width limit. When training with small learning rates, existing theory can describe the training dynamics of such networks. And, we present new theoretical results that explain the behavior at large learning rates. To summarize our findings, in the large width limit we identify two separate learning rate regimes, or phases, with the following characteristics.

**Lazy phase:** $\eta < 2/\lambda_0$ . For sufficiently small learning rate, the curvature $\lambda_t$ at training step $t$ remains constant throughout training, and the model becomes equivalent to a linear model (Jacot et al., 2018; Lee et al., 2019). The model converges to a nearby point in parameter space, and this behavior is sometimes called *lazy training* (Du et al., 2019; Zou et al., 2018; Allen-Zhu et al., 2019; Li & Liang, 2018; Chizat et al., 2019).

**Catapult phase:** $\eta_{\mathrm{crit}} < \eta < \eta_{\mathrm{max}}$ . At large learning rates the loss grows to be of order the width $n$ over a number of training steps that is of order $\log(n)$. During the same period, the curvature decreases until it is below $2/\eta$. Beyond this point the loss decreases and training converges, ultimately reaching a flat minimum (relative to that reached in the lazy phase). The gradient descent dynamics in this phase are visualized in Figure 1 and in Figure S1.

The maximum learning rate $\eta_{\mathrm{max}}$ (beyond which training no longer converges) depends on the setup. In our theoretical model the maximum learning rate is $\eta_{\mathrm{max}} = 4/\lambda_0$. For ReLU networks, we find empirically that $\eta_{\mathrm{max}} \approx 12/\lambda_0$.

### 1.3 LIMITATIONS

Our empirical results and our theoretical analysis focus on the case of MSE loss and on training with vanilla SGD, and do not extend to the case of cross-entropy loss, or to training with other optimizers such as momentum. Additionally, our theoretical analysis involves a 2-layer linear network, and does not apply to the case of networks with non-linearities such as ReLU.

### 1.4 RELATED WORKS

Our work builds on several existing results, which we now briefly review.

**The existing theory of infinite width networks is insufficient to describe large learning rates.**
A recent body of work has investigated the gradient descent dynamics of deep networks in the limit
of infinite width (Daniely, 2017; Jacot et al., 2018; Lee et al., 2019; Du et al., 2019; Zou et al.,
2018; Allen-Zhu et al., 2019; Li & Liang, 2018; Chizat et al., 2019; Mei et al., 2018; Rotskoff &
Vanden-Eijnden, 2018; Sirignano & Spiliopoulos, 2018; Woodworth et al., 2019; Naveh et al.; Xiao
et al., 2019). Of particular relevance is the work by Jacot et al. (2018) showing that gradient flow in
the space of functions is governed by a dynamical quantity called the Neural Tangent Kernel (NTK)
which is fixed at its initial value in this limit. Lee et al. (2019) showed this result is equivalent to
training the linearization of a model around its initialization in parameter space. Finally, moving
away from the strict limit of infinite width by working perturbatively, Dyer & Gur-Ari (2020); Huang
& Yau (2019) introduced an approach to computing the finite-width corrections to network evolution.

Despite this progress, in many practical deep learning settings, the neural network has finite width
and evolves nontrivially, with a large change in its associated Neural Tangent Kernel. Depending on
the architecture and hyperparameters, such networks may give superior performance. Prior work has
compared the performance of finite-width, SGD-trained deep networks with the infinite-width kernels
derived from the networks (Lee et al., 2018; Novak et al., 2019; Arora et al., 2019). Performance gaps
are observed in some cases, notably in convolutional networks, implying that existing infinite-width
theory is insufficient to explain the performance of deep networks in such settings where the network
evolves nontrivially.

**Large learning rate SGD improves generalization.**     SGD training with large initial learning rates
often leads to improved performance over training with small initial learning rates (see (Li et al.,
2019; Leclerc & Madry, 2020; Xie et al., 2020; Frankle et al., 2020; Jastrzebski et al., 2020) for
recent discussions). It has been suggested that one of the mechanisms underlying the benefit of
large learning rates is that noise from SGD leads to flat minima, and that flat minima generalize
better than sharp minima (Hochreiter & Schmidhuber, 1997; Keskar et al., 2016; Smith & Le, 2018;
Jiang et al., 2020; Park et al., 2019) (though see Dinh et al. (2017) for discussion of some caveats).
According to this suggestion, training with a large learning rate (or with a small batch size) can
improve performance because it leads to more stochasticity during training (Smith & Le, 2018; Mandt
et al., 2017; Smith et al., 2017; Smith et al., 2018).

We develop a connection between large learning rate and flatness of minima in models trained via
SGD. Unlike the relationship explored in most previous work though, this connection is not driven by
SGD noise, but arises solely as a result of training with a large initial learning rate, and holds even for
full batch gradient descent.

## 2    EXPERIMENTAL RESULTS

In a variety of deep learning settings, we find clear evidence of the different phases introduced
in Section 1. The experiments all use MSE loss, sufficiently wide networks, and vanilla SGD
with learning rate $\eta$. Parameters such as network architecture, choice of non-linearity, weight
parameterization, and regularization, do not significantly affect this conclusion.

In these experiments, we define the curvature $\lambda$ as the maximum eigenvalue of the Fisher Information
Matrix or, equivalently, as the maximum eigenvalue of the Neural Tangent Kernel (NTK). Given
a network function $f : \mathbb{R}^d \to \mathbb{R}$ with model parameters $\theta \in \mathbb{R}^p$, and a training set $\{(x_\alpha, y_\alpha)\}_{\alpha=1}^m$,
the NTK $\Theta : \mathbb{R}^d \times \mathbb{R}^d \to \mathbb{R}$ is defined by $\Theta(x, x') := \frac{1}{m} \sum_{\mu=1}^{p} \nabla_\theta f(x)^T \nabla_\theta f(x')$. We denote the
curvature at time $t$ by $\lambda_t$, equal to the maximum eigenvalue of $\Theta$. Another common measure of local
curvature is the maximum Hessian eigenvalue; at large width we expect these measures to agree
(Dyer & Gur-Ari, 2020), and we verify the agreement in Appendix D.6.

Building on the observed correlation between flat minima and generalization performance (Keskar
et al., 2016; Jiang et al., 2020), we conjecture that optimal performance occurs in the large learning
rate (catapult) phase, where optimization converges to a low curvature minimum. For a fixed
amount of computational budget, we find that this conjecture holds in all cases we tried. Even when
comparing different learning rates trained for a fixed amount of *physical time* $t_{\text{phys}} = t \cdot \eta$, we find
that performance of models trained in the catapult phase either matches or exceeds that of models
trained with learning rates below $\eta_{\text{crit}}$.

## 2.1 EARLY TIME CURVATURE DYNAMICS

Here we present empirical support for the lazy and catapult phases of training described in the previous section. Additional experimental results are presented in the appendix. We find that the lazy phase is characterized by small changes to the curvature and loss during training, while the catapult exhibits large deviations.

Figure 2 shows the curvature during the early part of training for two deep learning settings that involve sufficiently wide networks. The results are compared against the prediction of a phase transition at $\eta_{\text{crit}} = 2/\lambda_0$. For learning rates $\eta < \eta_{\text{crit}}$ (lazy phase), the curvature is independent of the learning rate and is approximately constant throughout training. For $\eta_{\text{crit}} < \eta < \eta_{\text{max}}$ we find that the curvature decreases during training to below $2/\eta$.

Figure 2 also shows the loss initially increasing before converging for large learning rates, a signature of the catapult effect. This transient behavior is very short, taking less than 10 steps to complete. Because of this, the training curve for the test loss is very similar and also shows the catapult effect. In these and other experiments involving ReLU networks, we find that $\eta_{\text{max}} \approx 12/\lambda_0$ is a good predictor of the maximum learning rate (in Appendix E.4 we discuss other nonlinearities). We conjecture that this is the typical maximum learning rate of networks with ReLU non-linearities.

## 2.2 GENERALIZATION PERFORMANCE

We now consider the performance of trained models in the different phases discussed in this work. Keskar et al. (2016) observed a correlation between the flatness of a minimum found by SGD and the generalization performance (see Jiang et al. (2020) for additional empirical confirmation of this correlation). In this work, we showed that the minima SGD finds are flatter in the catapult phase, as measured by the top kernel eigenvalue. Our measure of flatness differs from that of Keskar et al. (2016), but we expect that these measures to be correlated.

We therefore conjecture that optimal performance is often obtained for learning rates above $\eta_{\text{crit}}$ and below the maximum learning rate. In this section we test this conjecture empirically. We find that performance in the large learning rate regime always matches or exceeds the performance when $\eta < \eta_{\text{crit}}$. For a fixed compute budget, we find that the best performance is always found in the catapult phase.

Figure 3 shows the performance of a convolutional network and a Wide ResNet (WRN) trained on CIFAR-10. The experimental setup, which we now describe, was chosen to ensure a fair comparison of the performance across different learning rates. The network is trained with different initial learning rates, followed by a decay at a fixed physical time $t \cdot \eta$ to the same final learning rate. This schedule is introduced in order to ensure that all experiments have the same level of SGD noise toward the end of training.

We present results using two different stopping conditions. In Figure 3a, 3c, all models were trained for a fixed number of training steps. We find a significant performance gap between small and large learning rates, with the optimal learning rate above $\eta_{\text{crit}}$ and close to $\eta_{\text{max}}$. Beyond this learning rate, performance drops sharply.

The fixed compute stopping condition, while of practical interest, biases the results in favor of large learning rates. Indeed, in the limit of small learning rate, training for a fixed number of steps will keep the model close to initialization. To control for this, in Figure 3b,3d models were trained for the same amount of physical time $t \cdot \eta$. For the CNN of figure 3b, decaying the learning rate does not have a significant effect on performance and we observe that performance is flat up to $\eta_{\text{max}}$, and there is no correlation between our measure of curvature and generalization performance. Figure 3d shows the analogous experiment for WRN. When decaying the learning rate toward the end of training to control for SGD noise, we find that optimal performance is achieved above $\eta_{\text{crit}}$. In all these cases, $\eta_{\text{max}}$ is a good predictor of the maximal learning rate, despite significant differences in the architectures. Notice that by tuning the learning rate to the catapult phase, we are able to achieve performance using MSE loss, and without momentum, that is competitive with the best reported results for this model (Zagoruyko & Komodakis, 2016).

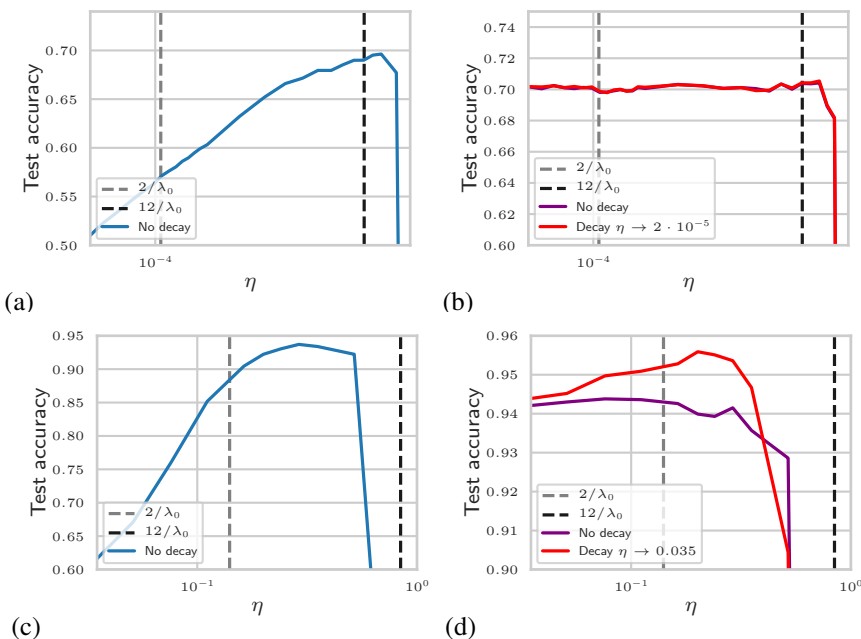

Figure 3: **Models perform best with a large learning rate.** Test accuracy vs learning rate for (a,b) a CNN trained on CIFAR-10 using SGD with batch size 256 and $L_2$ regularization ($\eta_{\text{crit}} \approx 10^{-4}$) and (c,d) WRN28-10 trained on CIFAR-10 using SGD with batch size 1024, $L_2$ regularization, and data augmentation ($\eta_{\text{crit}} \approx 0.14$); see Appendix A for details. (a,c) have a fixed compute budget: (a) 437k steps and (b) 12k steps. (b,d) have been evolved for a fixed amount of physical time: (b) was evolved for $475/\eta$ steps (purple) and evolved for 50k more steps at learning rate $2 \cdot 10^{-5}$ (red) and (d) was evolved for $3360/\eta$ steps with learning rate $\eta$ (purple) and then evolved for 4800 more steps at learning rate 0.035 (red). In all cases, optimal performance is achieved above $\eta_{\text{crit}}$ and close to the expected maximum learning rate, in agreement with our predictions.

In Appendix D.2, we present additional results for WRN on CIFAR-100, with similar conclusions. The fact that optimal performance happens in the catapult phase can also be observed for simple models like a fully-connected ReLU network trained on a subset of MNIST (see the Appendix).

## 3 GRADIENT DESCENT DYNAMICS OF WIDE, 2-LAYER LINEAR NETWORKS

We now turn to a theoretical analysis of the gradient descent dynamics of a two-layer linear network at large but finite width. While such a setting omits complexities such as depth and nonlinearity, our theoretical treatment already reveals the existence of three phases described in Section 1.2 with signatures that match our experiments.

Let the network function $f$ be given by $f(x) = n^{-1/2} v^T u x$. Here $n$ is the width (number of neurons in the hidden layer), $u, v \in \mathbb{R}^n$ are the model parameters (collectively denoted $\theta$), and $x \in \mathbb{R}$ is the training input. At initialization, the weights are drawn from $\mathcal{N}(0, 1)$. We prove the following

**Theorem 1.** *Consider a 2-layer linear network of width $n$, trained with MSE loss and learning rate $\eta$. The training data has a single sample with $(x, y) = (1, 0)$. Choose initial values $f_0 \neq 0$ and $\lambda_0 > 0$ for the function $f(x)$ and curvature $\lambda$. Let $\eta_{\text{crit}} := 2/\lambda_0$ and $\eta_{\text{max}} := 4/\lambda_0$, and choose $\delta > 0$.*

1. *Lazy phase: If $\eta < \eta_{\text{crit}}$ then gradient descent achieves loss $L < \delta$ in $\mathcal{O}(n^0)$ steps, and the final curvature $\lambda_f$ obeys $|\lambda_f - \lambda_0| = \mathcal{O}(n^{-1})$.*

2. *Catapult phase: If $\eta_{\text{crit}} < \eta < \eta_{\text{max}}$ then gradient descent achieves loss $L < \delta$, the final curvature obeys $\lambda_f \leq 2/\eta$, and during optimization the loss grows to be $\Omega(n \log^{-1}(n))$.*

3. *Divergent phase: If $\eta \geq \eta_{\text{max}}$ then gradient descent does not converge to a global minimum.*

The proof can be found in the appendix. We will complement the theorem with intuition about the dynamics of this model. In the appendix, we also generalize to the case of networks with arbitrary

input dimension. The gradient descent equations at training step $t$ are

$$u_{t+1} = u_t - \eta n^{-1/2} f_t v_t \,, \quad v_{t+1} = v_t - \eta n^{-1/2} f_t u_t \,. \tag{1}$$

The update equations in function space can be written in terms of the Neural Tangent Kernel. For this model, the kernel evaluated on the training set is a scalar which is equal to $\lambda$, its top eigenvalue, and is given by $\Theta(1,1) = \lambda = n^{-1} \left( \|v\|_2^2 + \|u\|_2^2 \right)$. At initialization, both $f^2$ and $\lambda$ scale as $n^0$ with the width $n$. The following update equations for $f$ and $\lambda$ at step $t$ can be derived from equation 1.

$$f_{t+1} = \left( 1 - \eta \lambda_t + \frac{\eta^2 f_t^2}{n} \right) f_t \,, \quad \lambda_{t+1} = \lambda_t + \frac{\eta f_t^2}{n} \left( \eta \lambda_t - 4 \right) \,. \tag{2}$$

It is important to note that these are the exact update equations for this model, and that no higher-order terms were neglected. We now analyze these dynamical equations assuming the width $n$ is large. Two learning rates that will be important in the analysis are $\eta_{\text{crit}} = 2/\lambda_0$ and $\eta_{\text{max}} = 4/\lambda_0$.

**Lazy phase.** Taking the strict infinite width limit, equations equation 2 become

$$f_{t+1} = (1 - \eta \lambda_t) f_t \,, \quad \lambda_{t+1} = \lambda_t \,. \tag{3}$$

When $\eta < \eta_{\text{crit}}$, $\lambda$ remains constant throughout training. This is a special case of NTK dynamics, where the kernel is constant and the network evolves as a linear model (Lee et al., 2019). The function and the loss both shrink to zero because the multiplicative factor obeys $|1 - \eta \lambda_t| < 1$. This convergence happens in $\mathcal{O}(n^0) = \mathcal{O}(1)$ steps.

**Catapult phase.** When $\eta_{\text{crit}} < \eta < \eta_{\text{max}}$, the loss diverges in the infinite width limit. Indeed, from equation 3 we see that the kernel is constant in the limit, while $f$ receives multiplicative updates where $|1 - \eta \lambda_t| > 1$. This is the well known instability of gradient descent dynamics for linear models with MSE loss. However, the underlying model is not linear in its parameters, and finite width contributions turn out to be important. We therefore relax the infinite width limit and analyze equations (2) for large but finite width, $n \gg 1$.

First, note that $\eta \lambda_0 - 4 < 0$ by assumption, and therefore the (additive) kernel updates are negative for all $t$. During early training, $|f_t|$ grows (as in the infinite width limit) while $\lambda_t$ remains constant up to small $\mathcal{O}(n^{-1})$ updates. After $t \sim \log(n)$ steps, $|f_t|$ grows to order $n^{1/2}$. At this point, the kernel updates are no longer negligible because $f_t^2/n$ is of order $n^0$. The kernel $\lambda_t$ receives negative, non-negligible updates while both $f_t$ and the loss continue to grow. This continues until the kernel is sufficiently small that the condition $\eta \lambda_t \lesssim 2$ is met.[2] We call this curvature-reduction effect the *catapult effect*. Beyond this point, $|1 - \eta \lambda_t| < 1$ holds, $|f_t|$ shrinks, and the loss converges to a global minimum. The $n$ dependence of the steps until optimization converges is $\log(n)$.

It is important for the analysis that we take a modified large width limit, in which the number of training steps grows like $\log(n)$ as $n$ becomes large. This is different than the large width limit commonly studied in the literature, in which the number of steps is kept fixed as the width is taken large. When using this modified limit, the analysis above holds even in the limit. Note as well that the catapult effect takes place over $\log(n)$ steps, and for practical networks will occur within the first 100 steps or so of training.

In the catapult phase, the kernel at the end of training is smaller by an order $n^0$ amount compared with its value at initialization. The kernel provides a local measure of the loss curvature. Therefore, the minima that SGD finds in the catapult phase are flatter than those it finds in the lazy phase. Contrast this situation, in which the kernel receives non-negligible updates, with the conclusions of Jacot et al. (2018) where the kernel is constant throughout training. The difference is due to the large learning rate, which leads to a breakdown of the linearized approximation even at large width.

Completing the analysis of this model, when $\eta > \eta_{\text{max}}$ the loss diverges because the kernel receives positive updates, accelerating the rate of growth of the function. Therefore, $\eta_{\text{max}} = 4/\lambda_0$ is the maximum learning rate of the model.

---

[2]The bound is not exact because of the term we neglected.

### 3.1 NON-PERTURBATIVE PHASE TRANSITION

The large width analysis of the small learning rate phase has been the subject of much work. In this phase, at infinite width, the network map evolves as a linear random features model, $f_{t+1}^{(0)} = f_t^{(0)} - \Theta f_t^{(0)}$, where $f^{(0)}$ is the function of the linearized model. At large but finite width (which we denote by $n$), corrections to this linear evolution can be systematically incorporated via a perturbative expansion (Taylor expansion) around infinite width Dyer & Gur-Ari (2020); Huang & Yau (2019), $f_t = f_t^{(0)} + \frac{1}{n} f_t^{(1)} + \cdots$. The evolution equations 3 of the solvable model are an example of this. At large width and in the small learning rate phase, the $O(n^{-1})$ terms are suppressed for all times. In contrast, the leading order dynamics of $f_t^{(0)}$ diverge when $\eta > \eta_{\text{crit}}$, and so the true evolution cannot be described by the linear model. Indeed, the logits grow to $\mathcal{O}(n^{1/2})$ and thus all terms in equation 3 are of the same order. Similarly, the growth observed empirically in the catapult phase for more general models cannot be described by truncating the perturbative series at any order, because the terms all become comparable.

## 4 DISCUSSION

Previous work (Jacot et al., 2018; Lee et al., 2019; Chizat et al., 2019) has studied the *lazy phase* of deep neural networks, which is known to occur for sufficiently small learning rates. In this work we argued that the lazy phase exists for learning rates smaller than $\eta_{\text{crit}} = 2/\lambda_0$, where $\lambda_0$ is the curvature at initialization. This critical learning rate corresponds to where a linear model, constructed from a deep network about its initialized parameters, would diverge under MSE loss. We pointed out the existence of the *catapult phase* in deep networks, corresponding to the learning rate regime $\eta_{\text{crit}} < \eta < \eta_{\text{max}}$. Its unique empirical signatures include the early-time growth of the loss and convergence to a flat minimum. Empirically, the existence and properties of the catapult phase can be observed across a variety network architectures and datasets. At yet larger learning rates $\eta > \eta_{\text{max}}$, SGD dynamics are unstable.

**A novel analysis illustrating the catapult phase.** Through our analytical treatment of a two-layer linear network, we are able to clarify the dynamics behind the catapult effect. Among these, we (i) derived the quantitative changes in loss and curvature and the time scales over which they occur; (ii) derived an expression for $\eta_{\text{max}}$ in terms of the curvature at initialization; (iii) specified the manner in which the lazy and catapult regimes are distinct *phases*, which we elaborate on below, in a novel modified infinite-width, infinite-time limit; and (iv) illustrated the dynamical mechanism stabilizing the catapult phase. Our approach reduces to analyzing two coupled difference equations relating the loss and curvature, which we hope may inspire a full treatment of deep networks with nonlinearities.

The change in behavior upon sweeping the learning rate from the lazy to catapult phase is reminiscent of phase transitions that commonly appear in physical systems such as ferromagnets or water, as one changes parameters such as temperature. Indeed, in Appendix C this connection is made concrete, with the change in behavior sharpening as width is increased. In particular, these transitions are non-perturbative: a Taylor series expansion of the linearized model that takes into account finite width corrections is not sufficient to describe the behavior beyond the critical learning rate.

**Catapult dynamics often improve generalization.** Our results shed light on the regularizing effect of training at large learning rates. The effect presented here is independent of the regularizing effect of stochastic gradient noise, which has been studied extensively. Building on previous works, we noted the observed correlation between flatness and generalization performance. Based on these observations, we expect the optimal performance to often occur for learning rates larger than $\eta_{\text{crit}}$, where the linearized model is unstable. Observing this effect required controlling for several confounding factors that affect the comparison of performance between different learning rates. Under a fair comparison, and also for a fixed compute budget, we find that this expectation holds in practice.

One outcome of our work is to address the performance gap between ordinary neural networks, and linear models inspired by the theory of infinite-width networks. Optimal performance is often obtained at large learning rates which are inaccessible to linearized models. In such cases, we expect the performance gap to persist even at arbitrarily large widths. We hope our work can further improve the understanding of deep learning dynamics and performance.

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

# A EXPERIMENTAL DETAILS

We are using JAX (Bradbury et al., 2018) and the Neural Tangents Library for our experiments (Novak et al., 2020).

All the models have been trained with Mean Squared Error normalized as $\mathcal{L}(\{x,y\}_B) = \frac{1}{2k|B|} \sum_{(x,y)\in B, i} (f^i(x) - y^i)^2$, where $k$ is the number of classes and $y^i$ are one-targets.

In a similar way, we have normalized the NTK as $\Theta_{ij}(x, x') = \frac{1}{k|B|} \sum_\alpha \partial_\alpha f^i(x) \partial_\alpha f^j(x')$ so that the eigenvalues of the NTK are the same as the non-zero eigenvalues of the Fisher information: $\frac{1}{k|B|} \sum_{x\in B, i} \partial_\alpha f^i(x) \partial_\beta f^i(x)$.

In our experiments we measure the top eigenvalue of the NTK using Lanczos' algorithm. We construct the NTK on a small batch of data, typically several hundred samples, compute the top eigenvalue, and then average over batches. In this work, we do not focus on precision aspects such as fluctuations in the top eigenvalue across batches.

All experiments that compare different learning rates use the same seed for the weights at initialization and we consider only one such initialization (unless otherwise stated) although we have not seen much variance in the phenomena described. We let $\sigma_w, \sigma_b$ denote the constant (width-independent) coefficient of the standard deviation of the weight and bias initializations, respectively.

Here we describe experimental settings specific to a figure.

**Figure 2a,2c,2e.** Fully connected, three hidden layers $w = 2048$, ReLU non-linearity trained using SGD (no momentum) on MNIST. Batch size= 512, using NTK normalization, $\sigma_w = \sqrt{2}, \sigma_b = 0$.

**Figures 2b,2d,2f.** Wide ResNet 28-18 trained on CIFAR10 with SGD (no momentum). Batch size of 128, LeCun initialization with $\sigma_w = \sqrt{2}, \sigma_b = 0, L_2 = 0$.

**Figures S4,S17** Fully connected network with one hidden layer and ReLU non-linearity trained on 512 samples of MNIST with SGD (no momentum). Batch size of 512, NTK initialization with $\sigma_w = \sqrt{2}, \sigma_b = 0$.

**Figures 3a,3b.** The convolutional network has the following architecture: $\text{Conv}_1(320) \rightarrow \text{ReLU} \rightarrow \text{Conv}_2(320) \rightarrow \text{ReLU} \rightarrow \text{MaxPool}((2,2), \text{'VALID'}) \rightarrow \text{Conv}_1(320) \rightarrow \text{ReLU} \rightarrow \text{Conv}_2(128) \rightarrow \text{MaxPool}((2,2), \text{'VALID'}) \rightarrow \text{Flatten}() \rightarrow \text{Dense}(256) \rightarrow \text{ReLU} \rightarrow \text{Dense}(10)$. $\text{Dense}(n)$ denotes a fully-connected layer with output dimension $n$. $\text{Conv}_1(n), \text{Conv}_2(n)$ denote convolutional layers with 'SAME' or 'VALID' padding and $n$ filters, respectively; all convolutional layers use $(3, 3)$ filters. $\text{MaxPool}((2,2), \text{'VALID'})$ performs max pooling with 'VALID' padding and a $(2,2)$ window size. LeCun initialization is used, with the standard deviation of the weights and biases drawn as $\sigma_w = \sqrt{2}$, $\sigma_b = 0.05$, respectively. Trained on CIFAR-10 with SGD, batch size of 256 and L2 regularization = 0.001.

**Figures 1, 3c,3d.** Wide ResNet on CIFAR10 using SGD (no momentum). Training on v3-8 TPUs with a total batch size of 1024 (and per device batch size of 128). They all use $L_2$ regularization= 0.0005, LeCun initialization with $\sigma_w = 1, \sigma_b = 0$. There is also data augmentation: we use flip, crop and mixup. With softmax classification, these models can get test accuracy of 0.965 if one uses cosine decay, so we don't observe a big performance decay due to using MSE. Furthermore, we are using JAX's implementation of Batch Norm which doesn't keep track of training batch statistics for test mode evaluation. We have not hyperparameter tuned for learning rates nor $L_2$ regularization parameter.

**Figures S5,S6.** Wide ResNet on CIFAR100 using SGD (no momentum). Same setting as figure 3c, 3d except for the different dataset, different L2 regularization = 0.000025 and label smoothing (we have subtracted 0.01 from the target one-hot labels).

**Figure S12.** Two hidden layer, ReLU network for one data point $x = 1, y = 1$.

**Figure S15.** Fully connected network with two hidden layers and tanh non-linearity trained on MNIST with SGD (no momentum). Batch size of 512, LeCun initialization with $\sigma_w = 1, \sigma_b = 0$.

**Figure S13a.** Two-hidden layer fully connected network trained on MNIST with batch size 512, NTK normalization with $\sigma_w = \sqrt{2}, \sigma_b = 0$. Trained using both momenta $\gamma = 0.9$ and vanilla SGD

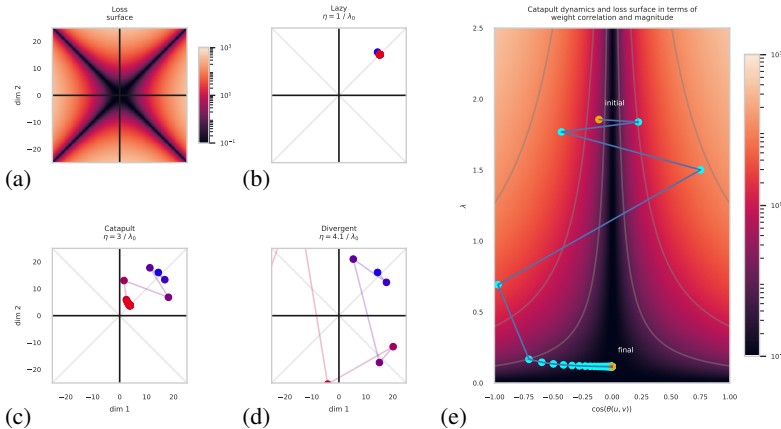

(a) (b)

(c) (d) (e)

Figure S1: Visualization of training dynamics in all three phases. In the **lazy phase**, the network is approximately linear in its parameters, and converges exponentially to a global minimum. In the **catapult phase**, the loss initially grows, while the weight norm and curvature decrease. Once the curvature is low enough, optimization converges. In the **divergent phase**, both the loss and parameter magnitudes diverge. *(a)-(d)* Loss surface and training dynamics visualized in a 2d linear subspace. The network has a single hidden layer with width $n = 500$, linear activations, and is trained with MSE loss on a single 1D sample $x = 1$ with label $y = 0$. The parameter subspace is defined by $u = [\text{dim1}]\, r + [\text{dim2}]\, s, v = [\text{dim1}]\, r - [\text{dim2}]\, s$, where $r$ and $s$ are orthonormal vectors, $u, v \in \mathbb{R}^n$ are the weight vectors, and $[\text{dim1}], [\text{dim2}]$ are the coordinates in the subspace. If initialized in this 2d subspace, $u_t$ and $v_t$ remain in the subspace throughout training, and so training dynamics can be fully visualized with a two dimensional plot. *(e)* Visualization of the loss surface and training dynamics in terms of a nonlinear reparameterization, providing interpretable properties: *x-axis* correlation between weight vectors, *y-axis* curvature $\lambda$. The trajectory shown is identical to that in (c), and in Figure 1.

for three different non-linearities: tanh, ReLU and identity (no non-linearity). The learning rate for each non-linearity was chosen to correspond to $\eta = \frac{1}{\lambda_0}$.

**Rest of appendix figures.** Small modifications of experiments in previous figures, specified in captions.

# B  THEORETICAL DETAILS

## B.1  THEOREM

For 1D input $x$ with label $y$, a network $f(x) = n^{-1/2}v^T u x$, learning rate $\eta$, and loss $L = (f(x) - y)^2/2$. We set $x = 1$ and $y = 0$. The curvature is $\lambda = n^{-1}(\|u\|^2 + \|v\|^2)$.

**Theorem 1.** *Consider a 2-layer linear network of width $n$, trained with MSE loss and learning rate $\eta$. The training data has a single sample with input $x = 1$ and label $y = 0$. Choose the initial values $f_0 \neq 0$ and $\lambda_0 > 0$ for the value of the function $f(x)$ and curvature $\lambda$. Let $\eta_{\mathrm{crit}} := 2/\lambda_0$ and $\eta_{\max} := 4/\lambda_0$, and choose $\delta > 0$.*

1. *Lazy phase: If $\eta < \eta_{\mathrm{crit}}$ then gradient descent achieves loss $L < \delta$ in $\mathcal{O}(n^0)$ steps, and the final curvature $\lambda_f$ obeys $|\lambda_f - \lambda_0| = \mathcal{O}(n^{-1})$.*

2. *Catapult phase: If $\eta_{\mathrm{crit}} < \eta < \eta_{\max}$ then gradient descent achieves loss $L < \delta$, the final curvature obeys $\lambda_f \leq 2/\eta$, and during optimization the loss grows to be $\Omega(n \log^{-1}(n))$.*

3. *Divergent phase: If $\eta \geq \eta_{\max}$ then gradient descent does not converge to a global minimum.*

*Proof.* Under gradient descent, the function and curvature update equations are given by

$$\Delta f_t = \left( -\eta\lambda_t + \frac{\eta^2 f_t^2}{n} \right) f_t, \quad \Delta\lambda_t = \frac{\eta f_t^2}{n}(\eta\lambda_t - 4). \tag{S1}$$

Here we denote $f = f(1)$, $\Delta f_t = f_{t+1} - f_t$, and $\Delta\lambda_t = \lambda_{t+1} - \lambda_t$.

**Convergence when $\eta < \eta_{\max}$.** If $\eta < \eta_{\max}$ then $\Delta\lambda_t \leq 0$ and $\lambda_t \leq \lambda_0$ for all $t$. Therefore,

$$\lambda_{T+1} = \lambda_0 + \sum_{t=0}^{T} \Delta\lambda_t \leq \lambda_0 + \frac{\eta}{n}(\eta\lambda_0 - 4)\sum_{t}^{T} f_t^2. \tag{S2}$$

Since $\lambda_{T+1} \geq 0$ we then have

$$\sum_{t}^{T} f_t^2 \leq \frac{n\lambda_0}{\eta(4 - \eta\lambda_0)}. \tag{S3}$$

From the monotone convergence theorem, the sum on the left-hand side converges when taking $T \to \infty$, and therefore $\lim_{t\to\infty} f_t = 0$, proving convergence when $\eta < \eta_{\max}$.

**Lazy phase.** Next, assume that $\eta < \eta_{\mathrm{crit}}$. We will first show that, when taking the infinite $n$ limit, optimization converges in a finite number of steps and the curvature is constant throughout optimization. We then show that the finite $n$ solution is close to the infinite $n$ one up to $O(n^{-1})$ corrections, which is enough to establish statement (1).

Taking $n \to \infty$, the update equations equation S1 become

$$\Delta f_t = -\eta\lambda_0 f_t, \quad \Delta\lambda_t = 0. \tag{S4}$$

Here we denote by $f_t^\infty$, $\lambda_t^\infty$ the function and curvature at step $t$ in the infinite width limit. For given ($n$ independent) initial conditions $f_0, \lambda_0$, in this limit the loss drops below $\delta$ in $O(n^0)$ steps, while the curvature remains constant.

Next, we compare the optimization trajectory $(f_t, \lambda_t)$ of the equations at finite $n$, with the trajectory $(f_t^\infty, \lambda_t^\infty)$ of the infinite $n$ equations. We set the initial conditions to be the same, $f_0 = f_0^\infty$ and $\lambda_0 = \lambda_0^\infty$. Next, let us show by induction on $t$ that the following holds at every step.

$$|f_t| = \mathcal{O}(n^0), \quad \lambda_t = \mathcal{O}(n^0), \quad |f_t - f_t^\infty| = \mathcal{O}(n^{-1}), \quad |\lambda_t - \lambda_t^\infty| = \mathcal{O}(n^{-1}). \tag{S5}$$

At $t = 0$ these relations hold by assumption. We now show the induction step. For the function trajectory,

$$
\begin{aligned}
|f_{t+1} - f_{t+1}^\infty| &= \left| \left( 1 - \eta\lambda_t + \frac{\eta^2 f_t^2}{n} \right) f_t - (1 - \eta\lambda_t^\infty) f_t^\infty \right| \\
&\leq |f_t - f_t^\infty| + \eta|\lambda_t^\infty f_t^\infty - \lambda_t f_t| + \frac{\eta^2}{n}|f_t^3| \\
&\leq |f_t - f_t^\infty| + \eta\lambda_t^\infty |f_t^\infty - f_t| + \eta|f_t||\lambda_t^\infty - \lambda_t| + \frac{\eta^2}{n}|f_t^3| \\
&= \mathcal{O}(n^{-1}).
\end{aligned}
\tag{S6}
$$

For the curvature trajectory,

$$
\begin{aligned}
|\lambda_{t+1} - \lambda_{t+1}^\infty| &= \left| \lambda_t + \frac{\eta f_t^2}{n}(\eta\lambda_t - 4) - \lambda_t^\infty \right| \\
&\leq |\lambda_t - \lambda_t^\infty| + \frac{\eta}{n}|f_t^2(\eta\lambda_t - 4)| \\
&= \mathcal{O}(n^{-1}).
\end{aligned}
\tag{S7}
$$

The remaining relations in equation S5 follow from the fact that $|f_t^\infty|$ and $\lambda_t^\infty$ are $\mathcal{O}(n^0)$. Since the infinite $n$ trajectory leads to $L < \delta$ in $\mathcal{O}(n^0)$ steps, this proves statement (1).

**Catapult phase.** We now show that any global minimum $(f, \lambda)$ with $f = 0$ and $\lambda > 2/\eta$ is a repulsive fixed point of the update equations. Assume that $\lambda_t = \frac{2}{\eta} + \delta$ for some $\delta > 0$, and that $0 < |f_t| < \epsilon$ for some $\epsilon$. For sufficiently small $\epsilon$,

$$
\left| \frac{f_{t+1}}{f_t} \right| = \left| 1 - \eta\lambda_t + \frac{\eta^2 f_t^2}{n} \right| = -1 + \eta\lambda_t - \frac{\eta^2 f_t^2}{n} > 1 + \delta\eta - \frac{\eta^2 \epsilon^2}{n} > 1.
\tag{S8}
$$

We see that $f_t$ grows in absolute value, proving that optimization does not converge. We showed that gradient descent converges in this case, and therefore the final curvature $\lambda_f < 2/\eta$.

It is left to show that the loss grows to be $\Omega(n \log^{-1}(n))$ during optimization. We choose $\epsilon > 0$. If it enough to show that if $\epsilon$ is sufficiently small, then $|f_t|$ grows larger than $\sqrt{\frac{n\epsilon}{\log(n)}}$ at a step $t = \mathcal{O}(\log(n))$.

From now on let us make the following assumptions. For some $\tau > 0$ (to be chosen),

$$
f_t^2 < \frac{n\epsilon}{\log(n)}, \quad 0 < t < \tau \log(n).
\tag{S9}
$$

Notice that these hold at initialization for sufficiently large $n$.

Using the fact that $\lambda_t \leq \lambda_0$ for all $t$, the curvature update is

$$
\Delta\lambda_t = \frac{\eta f_t^2}{n}(\eta\lambda_t - 4) < \frac{\eta}{n}\frac{n\epsilon}{\log(n)}(\eta\lambda_t - 4) \leq \frac{\eta\epsilon}{\log(n)}(\eta\lambda_0 - 4).
\tag{S10}
$$

With the assumptions above, we have

$$
|\lambda_t - \lambda_0| = \left| \sum_{t'=1}^{t} \Delta\lambda_{t'} \right| < \eta\tau\epsilon|\eta\lambda_0 - 4| = \mathcal{O}(\epsilon).
\tag{S11}
$$

We find that with the above assumptions, we have

$$
1 - \eta\lambda_t = 1 - \eta\lambda_0 + \eta|\lambda_t - \lambda_0| = 1 - \eta\lambda_0 + \mathcal{O}(\epsilon).
\tag{S12}
$$

Here we used again that $\lambda_t \leq \lambda_0$. In particular, since $1 - \eta\lambda_0 = -1 - 4\delta$ for some $\delta > 0$, then $1 - \eta\lambda_t < -1 - 2\delta$ for sufficiently small $\epsilon$.

Next, consider the $f_t$ update.

$$
\frac{f_{t+1}}{f_t} = 1 - \eta\lambda_t + \frac{\eta^2 f_t^2}{n} < -1 - 2\delta + \frac{\eta^2\epsilon}{\log(n)}.
\tag{S13}
$$

Again setting $\epsilon$ sufficiently small, we get $\left|\frac{f_{t+1}}{f_t}\right| > 1 + \delta$. We find that as long as the assumptions equation S9 hold,

$$|f_t| > (1+\delta)^t|f_0|. \tag{S14}$$

Let us now choose $\tau > \log^{-1}(1+\delta)$. Then for $t > \tau \log(n)/2$ and sufficiently small $\epsilon$,

$$|f_t| > n^{\tau \log(1+\delta)/2}|f_0| > \sqrt{n}|f_0| > \sqrt{\frac{n\epsilon}{\log(n)}}. \tag{S15}$$

This shows that the first assumption in equation S9 is the first to be violated when increasing $t$ from zero. Therefore, we find that $L_t > n\epsilon/\log(n) = \Omega(n/\log(n))$ at some point during training.

**Divergent phase.** Finally, suppose that $\eta \geq \eta_{\text{max}}$. Then $\Delta\lambda_t \geq 0$ and $\lambda_t \geq \lambda_0 > 2/\eta$ for all $t$. We showed that gradient descent can only converge to minima with $\lambda < 2/\eta$, and therefore gradient descent will not converge in this case. This concludes the proof of statement (3). $\qquad\square$

### B.2 EMPIRICAL PLOT OF SIMPLE MODEL

Figure S2 illustrates the dynamics in the catapult phase. For learning rates $\eta_{\text{crit}} < \eta < \eta_{\text{max}}$ we observe the catapult effect: the loss goes up before converging to zero. The curvature exhibits the expected sharp transitions as a function of the learning rate: it is constant in the lazy phase, decreases in the catapult phase, and diverges for $\eta > \eta_{\text{max}}$..

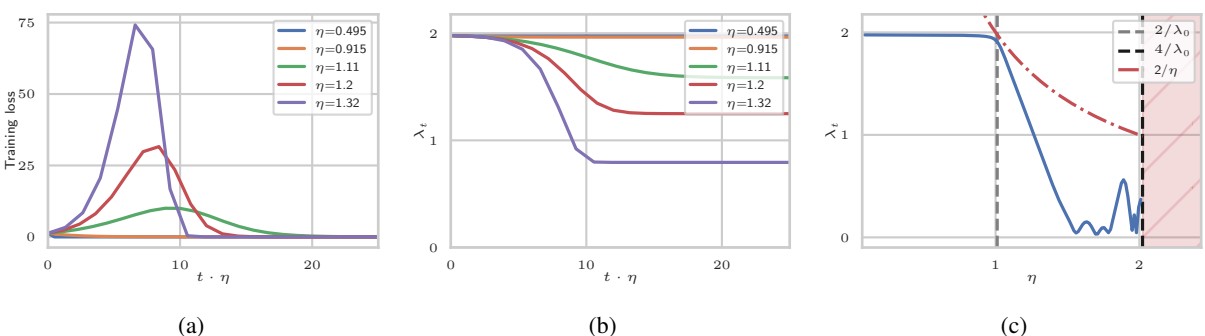

Figure S2: Empirical results for the gradient descent dynamics of the warmup model with $n = 10^3$, for which $\eta_{\text{crit}} \approx 1$. (a) Training loss for different learning rates. (b) Maximum NTK eigenvalue as a function of time. For $\eta > 1$, $\lambda_t$ decreases rapidly to a fixed value. (c) Maximum NTK eigenvalue at $t = 25/\eta$. The shaded area indicates learning rates for which training diverges empirically. The results are presented as a function of $t \cdot \eta$ (rather than $t$) for convenience.

### B.3 FULL MODEL ANALYSIS

Here we provide additional details on the theoretical analysis of the full model We introduce the notation $f_\alpha := f(x_\alpha)$ for the function evaluated on a training sample, $\tilde{f}_\alpha := f_\alpha - y_\alpha$ for the error, and $\Theta_{\alpha\beta} := \Theta(x_\alpha, x_\beta)$ for the kernel elements. We will treat $f, \tilde{f}$ evaluated on the training set as vectors in $\mathbb{R}^m$, whose elements are $f_\alpha, \tilde{f}_\alpha$. .The gradient descent update equations are

$$u_{ia}^{t+1} = u_{ia} - \frac{\eta}{\sqrt{nm}}v_i x_{a\alpha}\tilde{f}_\alpha, \quad v_i^{t+1} = v_i - \frac{\eta}{\sqrt{nm}}u_{ia}x_{a\alpha}\tilde{f}_\alpha. \tag{S16}$$

and

$$\Theta_{\alpha\beta} = \frac{1}{nm}(|v|^2 x_\alpha^T x_\beta + x_\alpha^T u^T u x_\beta) \tag{S17}$$

The update equations for the error and kernel evaluated on training set inputs are

$$\tilde{f}_\alpha^{t+1} = (\delta_{\alpha\beta} - \eta\Theta_{\alpha\beta})\tilde{f}_\beta + \frac{\eta^2}{nm}(x_\alpha^T\zeta)(f^T\tilde{f}), \tag{S18}$$

$$\Theta_{\alpha\beta}^{t+1} = \Theta_{\alpha\beta} - \frac{\eta}{nm}\left[(x_\beta^T\zeta)f_\alpha + (x_\alpha^T\zeta)f_\beta + \frac{2}{m}(x_\alpha^T x_\beta)(\tilde{f}^T f)\right]$$

$$+ \frac{\eta^2}{n^2 m}\left[|v|^2(x_\alpha^T\zeta)(x_\beta^T\zeta) + (\zeta^T u^T u\zeta)(x_\alpha^T x_\beta)\right]. \tag{S19}$$

Where $\zeta := \sum_\alpha \tilde{f}_\alpha x_\alpha/m \in \mathbb{R}^d$. We now consider the dynamics of the kernel projected onto the $\tilde{f}$ direction, which is given by

$$\tilde{f}^T\Theta_{t+1}\tilde{f} = \tilde{f}^T\Theta\tilde{f} + \frac{\eta}{n}\zeta^T\zeta\left(\eta\tilde{f}^T\Theta\tilde{f} - 4f^T\tilde{f}\right). \tag{S20}$$

Let us now analyze the phase structure of equation S18 and equation S20. For now, we neglect the last term on the right-hand side of equation S18 (at initialization this term is of order $n^{-1}$ and is negligible at large width). Let $\lambda_0$ be the maximal eigenvalue of the kernel at initialization, and let $e^{\max} \in \mathbb{R}^m$ be the corresponding eigenvector. Notice that $\tilde{f}$ projected onto the top eigenvector evolves as

$$(e^{\max})^T\tilde{f}_{t+1} = (1 - \eta\lambda)e^{\max T}\tilde{f} + \mathcal{O}(n^{-1}). \tag{S21}$$

**Lazy phase.** When $\eta\lambda_0 < 2$, we see that $|e^{\max T}\tilde{f}^t|$ shrinks during training. The kernel updates are of order $n^{-1}$, while convergence happens in order $n^0$ steps. Therefore the kernel does not change by much during training. This is a special case of the NTK result (Jacot et al., 2018). Effectively, the model evolves as a linear model in this phase.

**Catapult phase.** When $2 < \eta\lambda_0 < 4$, $\|\tilde{f}\|_2$ grows exponentially fast, and it grows fastest in the $e^{\max}$ direction. Therefore, the vector $\tilde{f}$ becomes aligned with $e^{\max}$ after a number of steps that is of order $n^0$. Also, $f$ itself grows quickly while the label is constant, and so we find that $f \approx \tilde{f} \approx (e^{\max T}\tilde{f})e^{\max}$ after a similar number of steps. When these approximations hold, notice that $\tilde{f}^T\Theta\tilde{f} \approx \lambda \cdot \|\tilde{f}\|_2^2$.

Regarding the evolution of the top eigenvector itself, it is easy to see from equation S19 that the kernel updates preserve the two separate subspaces of eigenvectors with $\lambda_i < 2/\eta$ and $\lambda_i > 2/\eta$. If there is only one vector with $\eta\lambda_i > 2$, then the $e^{\max}$ direction stays constant during training. If there are multiple such vectors, we expect that ones with larger eigenvalues grow exponentially faster, and thus we expect that the eigenvectors are approximately preserved.

Now, from equation equation S20 we can then derive an approximate equation for the evolution of the top NTK eigenvalue.

$$\lambda_{t+1} \approx \lambda + \frac{\eta}{n}\zeta^T\zeta(\eta\lambda - 4). \tag{S22}$$

While $\tilde{f}$ grows exponentially fast, so will $\zeta$. When $\zeta_t$ becomes of order $n^{1/2}$, the updates to the top eigenvalue become of order $n^0$ (and negative), causing $\lambda_t$ to decrease by a non-negligible amount. This will continue until $\lambda_t < 2/\eta$, at which point $\tilde{f}_t$ will start converging to zero. Eventually, after a number of steps of order $\log(n)$, gradient descent will converge to a global minimum that has a lower curvature than the curvature at initialization.

The justification for dropping the order $n^{-1}$ term in equation S21 was explained in the warmup model: While this term may affect the details of the dynamics, eventually the maximum kernel eigenvalue must drop below $2/\eta$ for the component $e^{\max T}\tilde{f}$ of the error (and therefore for the loss) to converge to zero.

**Divergent phase.** When $\eta\lambda_0 > 4$, both $\|\tilde{f}\|_2^2$ and $\lambda$ will grow, and optimization will diverge. Therefore, $4/\lambda_0$ is the maximum learning rate for this model.

## C    MODEL DYNAMICS CLOSE TO THE CRITICAL LEARNING RATE

Here we consider the gradient descent dynamics of the model analyzed in Section 3, for learning rates $\eta$ that are close to the critical point $\eta_{\text{crit}} = 2/\lambda_0$. The analysis reveals that the gradient descent dynamics of the model are qualitatively different above and below this point. For example, the loss decreases monotonically during training when $\eta < \eta_{\text{crit}}$, but not when $\eta > \eta_{\text{crit}}$. In this section we show that the transition from small to large learning rate becomes sharp once we take the modified large width limit, in the following sense: certain functions of the learning rate become non-analytic at $\eta_{\text{crit}}$ in the limit. This sharp transition bears close resemblance to phase transitions of the kind found in physical systems, such as the transition between the liquid and gaseous phases of water. In particular, our case involves a dynamical system, where the dynamics are governed by the gradient descent equations. These dynamics undergo a phase transition as a function of the learning rate — an external parameter. We point to the logistic map May (1976) as a well-known example of a dynamical system that undergoes phase transitions as a function of an external parameter.

### C.1    NON-PERTURBATIVE DYNAMICS

A phase transition is a drastic change in a system's behavior incurred under a small change in external parameters. Mathematically, it is a non-analyticity in some property of the system as a function of these parameters. For example, consider the property $\lambda_*(\eta)$, the curvature of the model at the end of training as a function of the learning rate. In the modified large width limit, $\lambda_*(\eta)$ is constant for $\eta < \eta_{\text{crit}}$, but not for $\eta > \eta_{\text{crit}}$. Therefore, this function is not analytic at $\eta_{\text{crit}}$. Notice that this statement is true in the limit but not necessarily at finite width, where the final curvature may be an analytic function of the learning rate even at $\eta_{\text{crit}}$. It is well known in physics that phase transitions only occur in a limit where the number of dynamical variables (in this case the number of model parameters) is taken to infinity. One immediate consequence of the non-analyticity at $\eta_{\text{crit}}$ is that the large learning rate phase is inaccessible from the small learning rate phase via a perturbative expansion. In other words, we cannot describe all properties of the model for some $\eta > \eta_{\text{crit}}$ by doing a Taylor expansion around a point $\eta_0 < \eta_{\text{crit}}$ and keeping a finite number of terms.

Dyer & Gur-Ari (2020); Huang & Yau (2019) developed a formalism that allows one to compute finite-width corrections to various properties of deep networks, using a perturbative expansion around the infinite width limit. We have argued that the usual infinite width approximation to the training dynamics is not valid for learning rates above $\eta_{\text{crit}}$, and that a full analysis must account for large finite-width effects. One may have hoped that including the perturbative finite-width corrections discussed in Dyer & Gur-Ari (2020); Huang & Yau (2019) would allow us to regain analytic control over the dynamics. The results presented here suggest that this is not the case: For $\eta > \eta_{\text{crit}}$, we expect that the perturbative expansion will not provide a good approximation to the gradient descent dynamics at any finite order in inverse width.

### C.2    CRITICAL EXPONENTS

When the external parameters are close to a phase transition, one often finds that the dynamical properties of the system obey power law behavior. The exponents of these power laws (called *critical exponents*) are of interest because they are often found to be universal, in the sense that the same set of exponents is often found to describe the phase transitions of completely different physical systems.

Here we consider $t_*(\eta)$, the number of steps until convergence, as a function of the learning rate. We will now show that $t_*$ exhibits power-law behavior when $\eta$ is close to $\eta_{\text{crit}}$. For simplicity we consider the warmup model studied in Section 3. First, suppose that we are below the transition, setting $\eta\lambda_0 = 2 - \epsilon$ for some small $\epsilon > 0$. From the update equation, $f_{t+1} \approx (1 - \eta\lambda_t)f_t \approx -(1 - \epsilon)f_t$ we see that $f_t$ will converge to some fixed small value $f_*$ after time $t_* \approx \epsilon^{-1}\log(f_*^{-1}) \sim \epsilon^{-1}$. Here we assumed that $\lambda_t$ is constant in $t$, which is true as long as $t_*$ is independent of $n$ (namely we fix $\epsilon$ and then take $n$ large). Therefore, the convergence time below the transition scales as $t_* \sim (\eta_{\text{crit}} - \eta)^{-1}$, and the critical exponent is -1.

Next, suppose that $\eta\lambda_0 = 2 + \epsilon$ with $\epsilon > 0$. Now the update equation reads $f_{t+1} \approx -(1 + \epsilon)f_t$. This approximation holds early during training, when the curvature updates are small. Initially, $|f_t|$ will grow until it is of order $\sqrt{n}$, at which point the updates to $\lambda_t$ become of order $n^0$. This will happen in time $\hat{t} \sim \epsilon^{-1}\log\sqrt{n}$. Following this, the optimizer will converge. At this point $\eta\lambda_t$ is no longer

tuned to be close to the transition, and so the convergence time measured from this point on will not be sensitive to $\epsilon$. Therefore, for small $\epsilon$ the convergence time will be dominated by the early part of training, namely $t_* \approx \hat{t} \sim \epsilon^{-1}$. The critical exponent is again -1. Figure S3 show an empirical verification of this behavior.

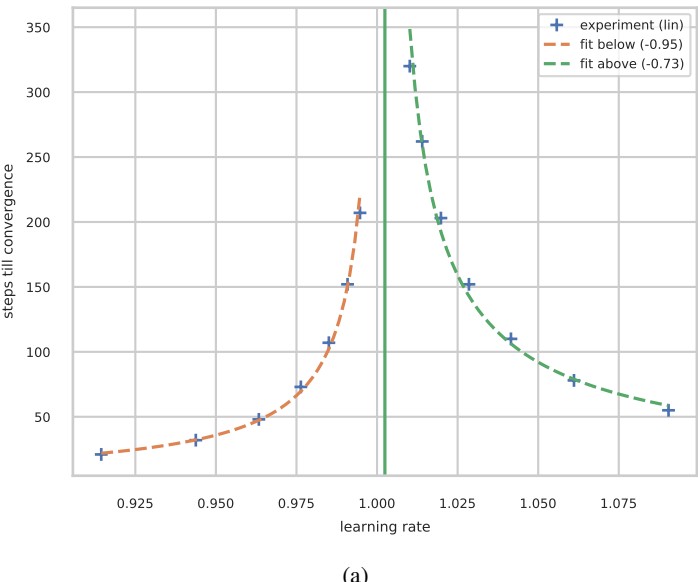

(a)

Figure S3: The convergence time diverges when the learning rate is close to the critical value $\eta_{\mathrm{crit}}$, indicated by the solid green line. The measured exponents (shown in parentheses) are close to the predicted value of -1. Experiment involves the warmup model of Section 3 with width 16,000.

## D    EXPERIMENTAL RESULTS: LATE TIME PERFORMANCE

### D.1    SIMPLE MNIST EXPERIMENT

Figure S4 shows the accuracy as a function of the learning rate for a fully-connected ReLU network trained on a subset of MNIST. We find that the optimal performance is achieved above $\eta_{\mathrm{crit}}$ and close to $\eta_{\mathrm{max}} = 12/\lambda_0$, the expected maximum learning rate.

### D.2    CIFAR-100 PERFORMANCE

We can also repeat the performance experiments for CIFAR-100 and the same Wide ResNet 28-10 setup. In this case, using MSE and SGD we require to evolve the system for longer times, which requires a smaller $L_2$ regularization. We didn't tune for it, but found that $2.5 \times 10^{-5}$ works. With only one decay we can get within $3\%$ of the Zagoruyko & Komodakis (2016) performance that used softmax classification and two learning rate decays. However, evolution for longer time is needed: we found that different learning rates converge at $\approx 2000$ physical epochs. Similar to the main text experiments, we observe that if we decay after evolving for the same amount of physical epochs, larger learning rates do better. See figure S5.

### D.3    DIFFERENT LEARNING RATES CONVERGE AT THE SAME PHYSICAL TIME

We can also plot the test accuracy versus physical time for different learning rates to show that for vanilla SGD, the performance curves of different learning rates are basically on top of each other if we plot them in physical time, which is why we find that the fair comparison between learning rates should be at the same physical time.

We have picked a subset of learning rates of the previous WRN28-18 CIFAR100 experiment of Appendix D.2. In figure S6, we see how even if the curves are slightly different they converge to

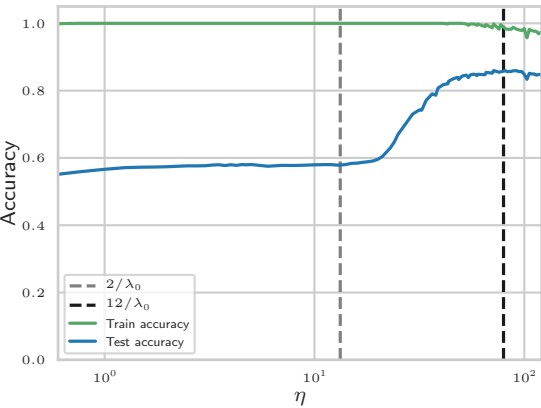

Figure S4: Final accuracy versus learning rate for a fully-connected 1 hidden layer ReLU network, trained on 512 samples of MNIST with full-batch gradient descent until training accuracy reaches 1 or 700k physical steps (see Appendix A for details). We used a subset of samples to accentuate the performance difference between phases. The optimal performance is obtained when the learning rate is above $\eta_{\text{crit}}$, and close to $\eta_{\text{max}}$.

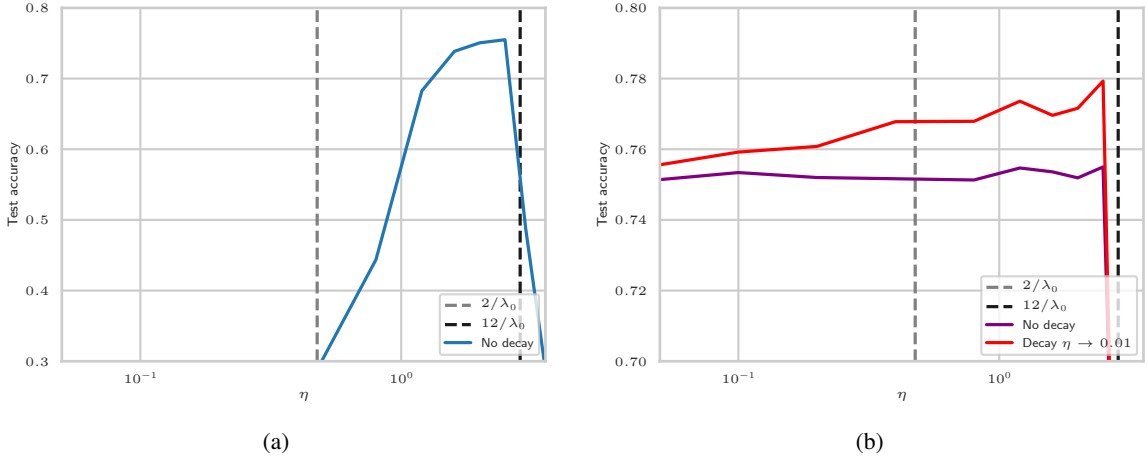

| (a) | (b) |

Figure S5: Test accuracy vs learning rate for WRN28-10 and CIFAR100 with vanilla SGD, $L_2$ regularization, data augmentation, label smoothing and batch size 1024. The critical learning rate is $\eta_{\text{crit}} \approx 0.4$. (a) Evolved for 38400 steps. (b) Evolved for $96000/\eta$ steps with learning rate $\eta$ (blue) and then evolved for 7200 more steps at learning rate 0.01 (red).

roughly the same accuracy. The only curve which is slightly different is $\eta = 2.5$ which is a rather high learning rate (close to $\frac{12}{\lambda_0}$).

### D.4 COMPARISON OF LEARNING RATES FOR DIFFERENT $L_2$ REGULARIZATION FOR WRN28-10 ON CIFAR10

Even if in the main section we have considered a model with fixed $L_2$ regularization, we can study the effect without $L_2$ or with a different value. In these two examples, we will be considering the same setup as figures 3c,3d.

Without $L_2$ regularization, we see that the larger learning rate does better even in the absence of learning rate decay, although training takes a really long time. In our experience, comparing this setup with state of the art, $L_2 = 0$ regularization makes the experiment take longer before convergence but does not influence performance much.

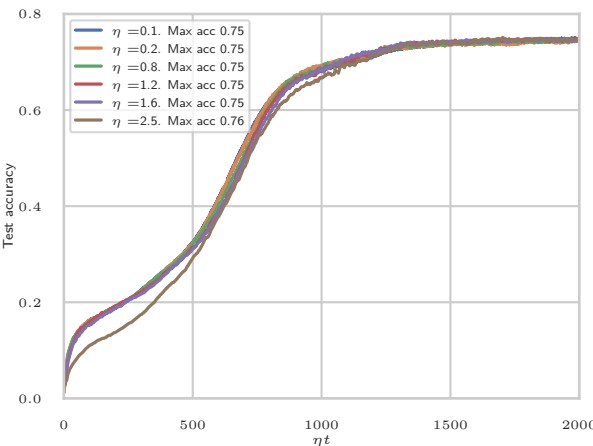

Figure S6: Test accuracy vs physical time for different learning rates in the WRN CIFAR100 experiment of the previous section D.2

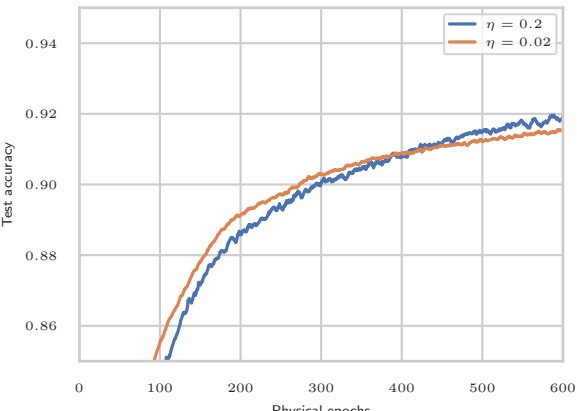

Figure S7: WRN28-10 on CIFAR10 without $L_2$. Same setup as 3d but evolved for longer times.

In the presence of $L_2$ regularization we picked the particular value $L_2 = 0.0005$ in order to make sure that our conclusion is not dependent on the choice of $L_2$, the only hyperparameter (other than $\eta$), we have considered a larger $L_2 = 0.001$. We see that the optimal performance in physical time is also peaked in the catapult phase, although the difference here is smaller.

## D.5 TRAINING ACCURACY PLOTS

The training accuracies of the previous experiments are shown in figure S9.

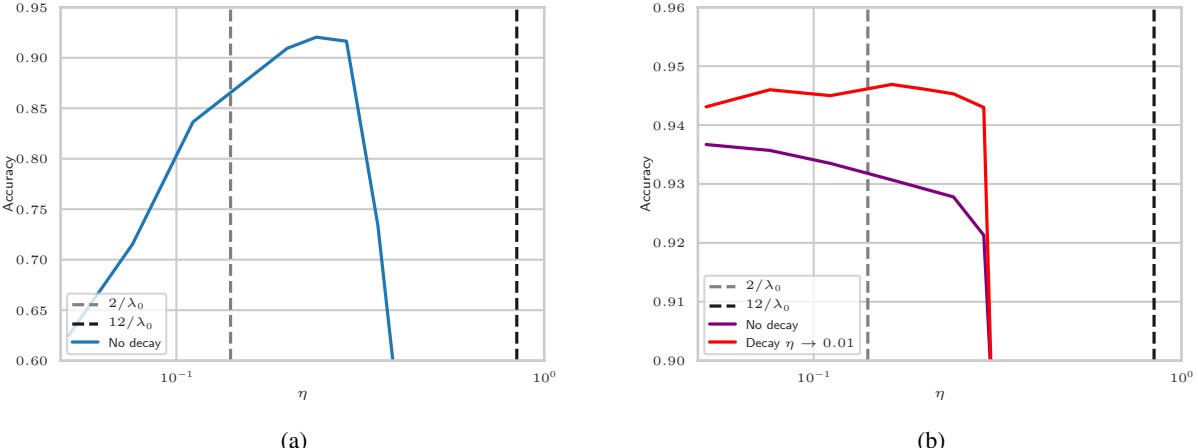

(a)                                                              (b)

Figure S8: Test accuracies for a larger $L_2$ CIFAR10 experiment like that of the main section. (a) WRN CIFAR-10 7200 steps as in figure 3c. (b) WRN CIFAR10 2400 physical steps and then 4800 more steps at learning rate 0.01 as in figure 3d.

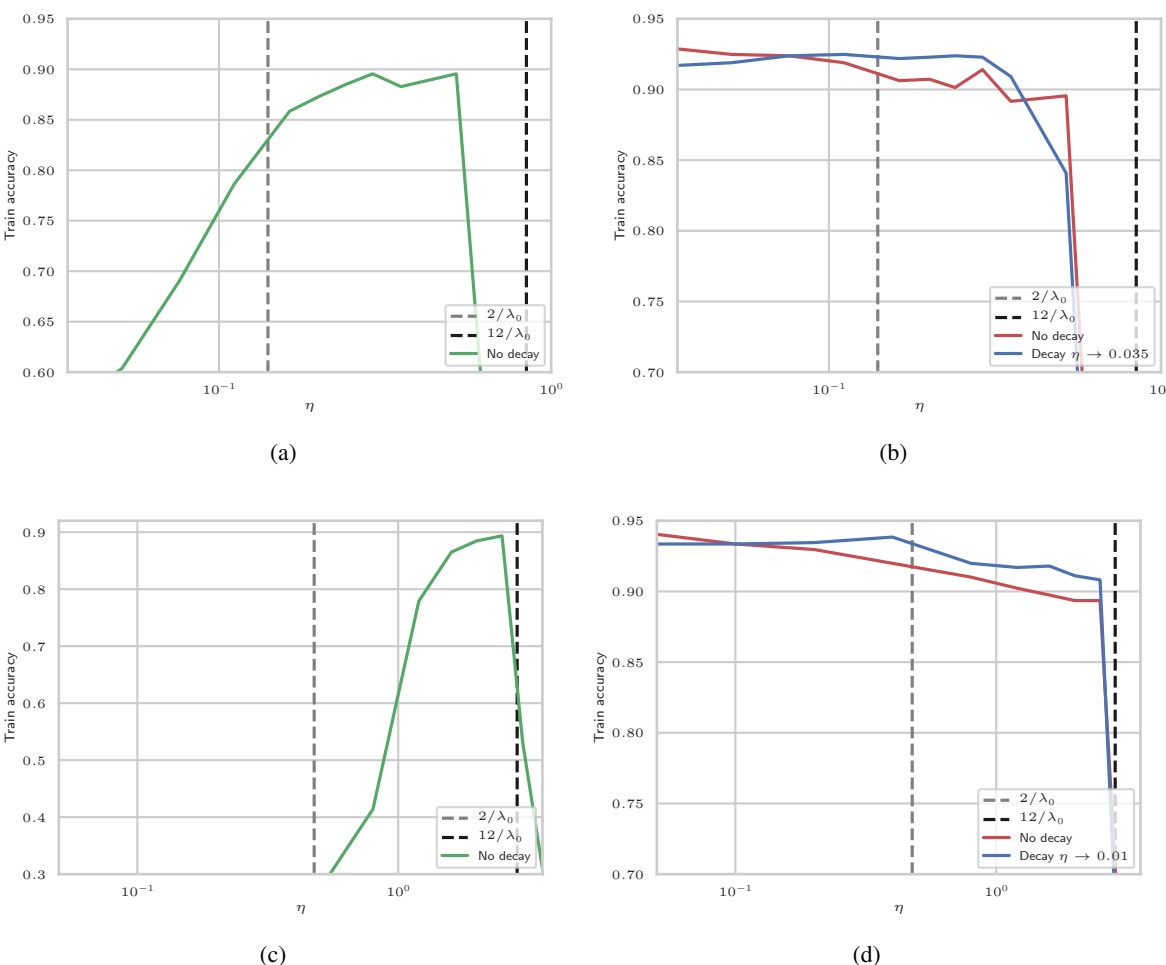

Figure S9: Training accuracies for the performance experiments. Smaller learning rates have higher training accuracy when compared in physical time. However, they still perform worse for a fixed number of steps. (a) WRN CIFAR-10 12000 steps as in figure 3c. (b) WRN CIFAR10 3360 physical steps as in figure 3d. (c) WRN CIFAR100 38400 steps as in figure S5a.(d) WRN CIFAR100 96000 physical steps as in figure S5b.

## D.6 EQUIVALENCE OF $\lambda_0$ AND THE HESSIAN EIGENVALUE

In figure S10 we run an experiment for a shallow Wide Resnet with 10 layers (1 block), without data augmentation and measure of the maximum eigenvalues of the NTK and Hessian for different widths. We see that they track each other fairly well for all times. There is intrinsic noise because different samples are used to compute these eigenvalues.

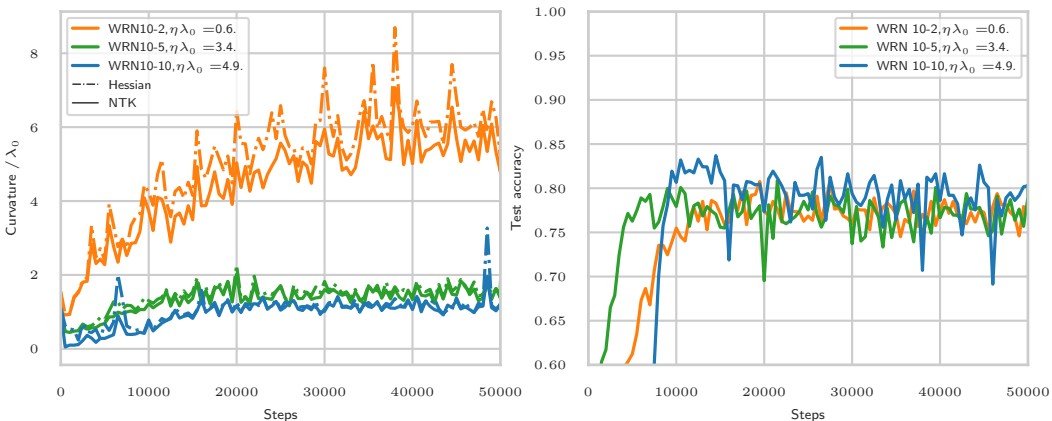

Figure S10: NTK vs Hessian eigenvalues for shallow Wide Resnet trained on CIFAR10 during training. These values are normalized by $\lambda_0$

## D.7 LATE TIME BEHAVIOUR OF EIGENVALUES

We can compute the eigenvalues at late times for the experiment of figure 3d . Figure S11 shows $\lambda$ at the final step of training (before decaying). We see that the dependence of the curvature on the learning rate is preserved at late times.

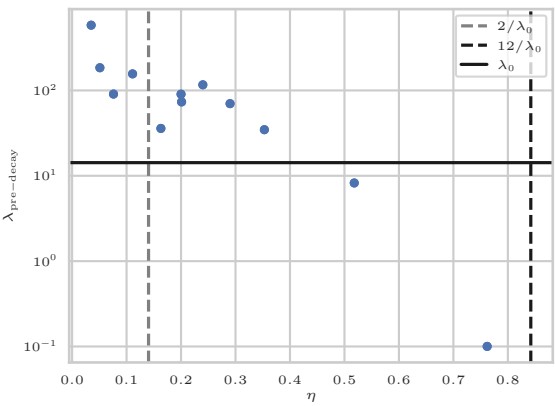

Figure S11: $\lambda$ vs $\eta$ at late times for the experiment of figure 3d.

# E EXPERIMENTAL RESULTS: EARLY TIME DYNAMICS

## E.1 ReLU ACTIVATIONS FOR THE SIMPLE MODEL

In the main text we have been using ReLU non-linearities. Compared with the simple model with no non-linearities, ReLU networks have a broader trainability regime after $\eta = \frac{4}{\lambda_0}$. It looks like these networks generically will train until $\eta = \frac{12}{\lambda_0}$. This is a generic feature of deep ReLU networks and can be already observed for a non-linear generalization of the single sample model in Section 3 with a target $y = 1$, two hidden layers and a ReLU non-linearity: $f = u.ReLU(w.ReLU(v))$, as shown in figure S12). In this single sample context for $\eta \geq \frac{12}{\lambda}$, the loss doesn't diverge but the neurons die and end up giving the trivial $f = 0$ function. For deep networks with more than one hidden layer and multiple samples, as discussed in the main text, we observe that the loss diverges after $\sim \frac{12}{\lambda}$.

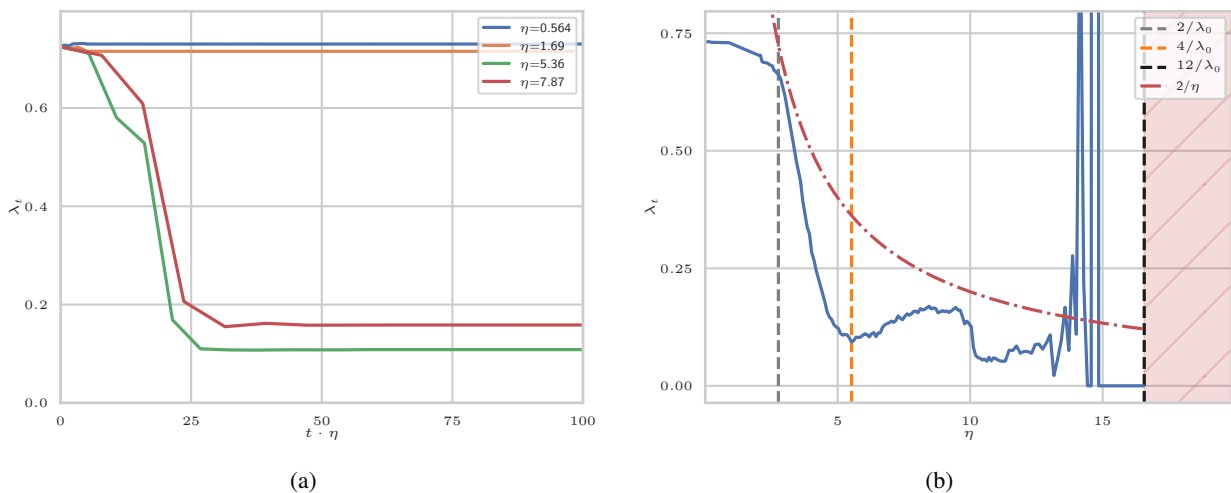

(a)  (b)

Figure S12: Simple model ReLU non-linearity ($\eta_{\text{crit}} = 2.54$). (b) is evaluated at physical time 100.

## E.2 MOMENTA

The effect of the optimizer also affects these dynamics. If we consider a similar setup with momenta, first we expect that a linear model converges in a broader range $\eta < \frac{2}{\lambda_0}(1 + \gamma)$. For smooth non-linearities, we observe that for $\eta < \frac{2}{\lambda_0}$, the $\lambda_t$ is constant. However this is not true for ReLU, see figure S13a. In fact, for ReLu networks, we observe that there is a small learning rate, roughly $\eta_{\text{eff,crit}} = \frac{\eta_{\text{crit}}}{1-\gamma}$, below which the time dynamics of $\lambda_t$ is similar (but non-constant). However, for $\eta > \eta_{\text{eff,crit}}$, there are strong time dynamics, we illustrate this in figure S13b with a 3 hidden layer ReLu network.

## E.3 EFFECT OF $L_2$ REGULARIZATION TO EARLY TIME DYNAMICS

We don't expect $L_2$ regularization to affect the early time dynamics, but because of the strong rearrangement that goes on in the first steps, it could potentially have a non-trivial effect; among other things, the Hessian spectrum necessarily is decaying. We can see how the dynamics that drives the rearrangement is roughly the same, even in the maximum eigenvalue at early times is decreasing slowly.

## E.4 TANH ACTIVATIONS

We observe that for Tanh activation, $\eta_{\text{max}}$ is closer to the simple model expectation $\frac{4}{\lambda_0}$, see figure S15.

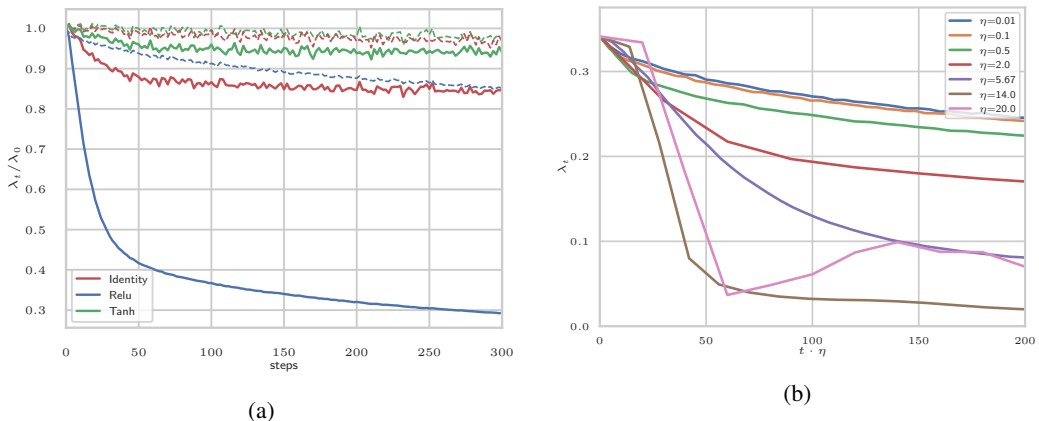

(a)                  (b)

Figure S13: (a) Evolution of the normalized curvature $\lambda_t/\lambda_0$ for $d = 2$ $w = 2048$ FC connected networks evolved with momenta (same networks with SGD with dashed line for reference) evolved for $\eta = \frac{1}{\lambda_0}$. We observe that ReLU networks evolved with momenta doesn't have a constant kernel in the naive 'lazy' phase. (b) $\eta_{\text{crit}} = 6.96$, $\eta_{\text{crit,eff}} = 0.69$ Same setup as the FC network of figure 2 with momenta $\gamma = 0.9$: fully connected, three hidden layers $w = 2048$, ReLU non-linearity. $\eta_{\text{crit}}$ is slightly different due to variations at initialization.

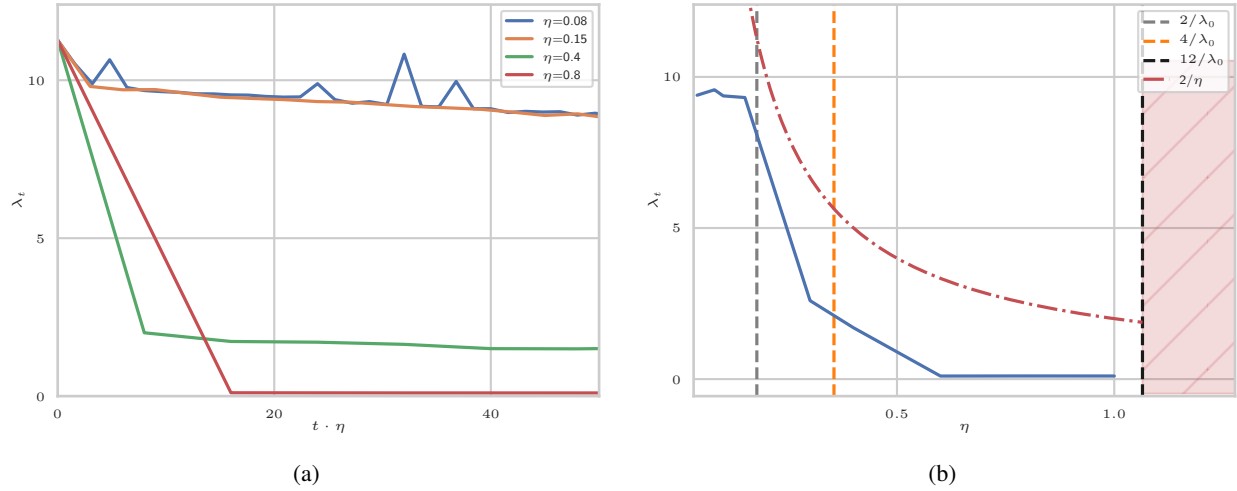

(a)                  (b)

Figure S14: Same WRN as figure 2d,f with $L_2$ regularization= 0.0005. Dynamics in physical steps of the $\lambda_t$ and $\lambda_t$ vs $\eta$. $\eta_{\text{crit}} = 0.18$ a) $\lambda_t$, b) $\lambda_t$ at physical time 25

### E.5 WRN NTK NORMALIZATION

As illustrated in the text in figures 2c, 2e we also see this behaviour for NTK normalization. For completeness we include the WRN model with NTK normalization. From the linearized intuition, we expect the phases to also be determined by the quantity $\eta\lambda_t$, independently of the normalization. Figure S16 has the same setup as in figure 2.

## F RESTORATION OF LINEAR DYNAMICS

One striking prediction of the the catapult mechanism is that after a period of excursion, the logit differences settle back to $\mathcal{O}(1)$ values, the NTK stops changing, and evolution is again well approximated by a linear model with constant kernel at large width.

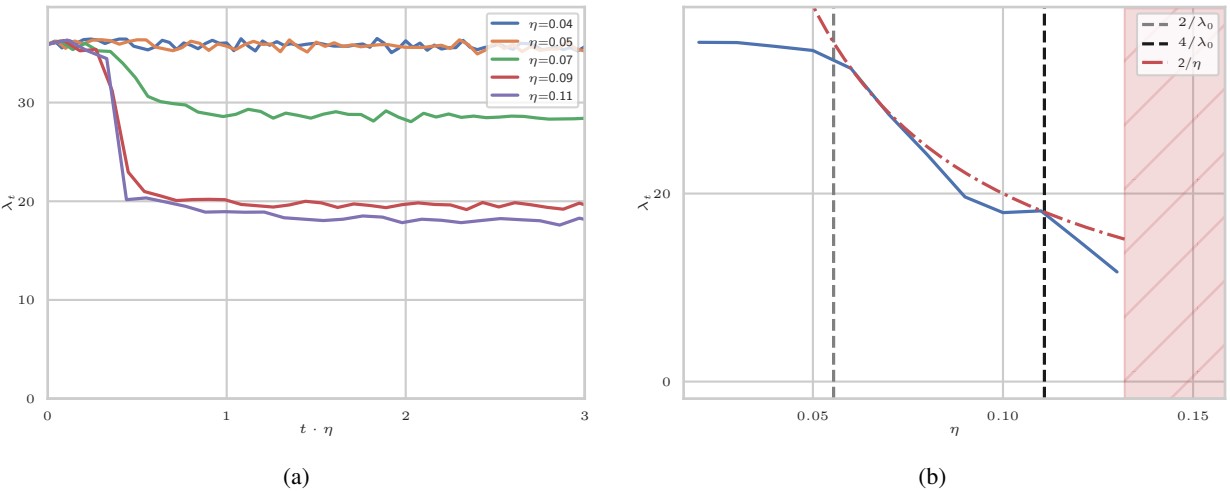

Figure S15: Maximum NTK eigenvalue $\lambda$ at early times for a 2 hidden layer fully connected network with tanh non-linearity trained on MNIST, with $\eta_{\text{crit}} = 0.06$. (a) Early time dynamics of the curvature for learning rates in the linear and catapult phase. (b) $\lambda$ measured at $\eta t = 3$.

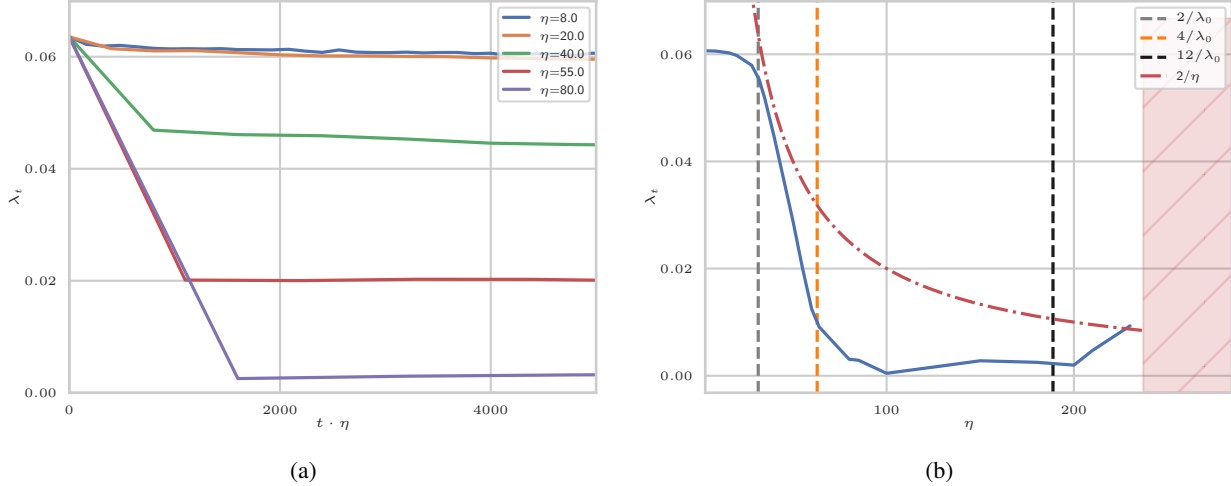

Figure S16: Same as Figures 2d,2f but with NTK normalization. a,b) Wide Resnet 28-10. $\eta_{\text{crit}} = 31.47, \lambda$ vs $\eta$ at physical time 4000

We speculate that the return to linearity and constancy of the kernel may hold asymptotically in width generally for a range of learning rates above $\eta_{\text{crit}}$. We test this by evolving the model for order $\log(n)$ steps until the catapult effect is over, linearizing the model, and comparing the evolution of the two models beyond this point. Figure S17 shows an example of this. At fixed width, the accuracy of the linear and non-linear networks match for a range of learning rates above the transition up to $4/\lambda_0$. We present additional evidence for this asymptotic linearization behavior in the Supplement.

## F.1 ADDITIONAL EVIDENCE FOR LINEARIZATION IN THE CATAPULT PHASE.

Here we present some more detailed evidence for the re-emergence of linear dynamics in the catapult phase. Figure S18 show results for models trained on subsets of MNIST with learning rates $\eta > \eta_{\text{crit}}$. In figure Figure S18a we see that for a one-hidden-layer fully connected model trained on 512 MNIST images, the performance of the full non-linear model and model linearized after 10 steps track closely.

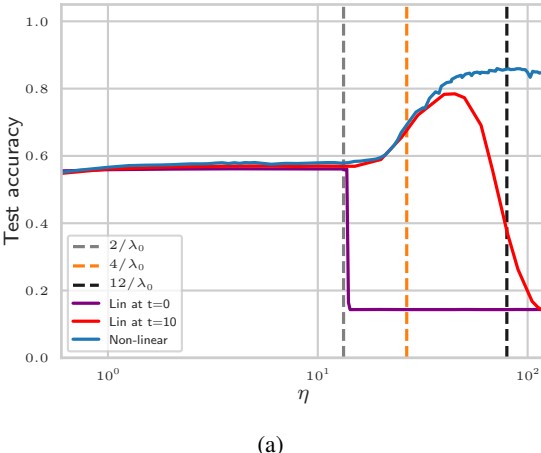

(a)

Figure S17: Evidence for linear dynamics after the catapult effect is over. Here we show the same model as in Figure S4 with the addition of models linearized at step 0 and another linearized at step 10. We observe that the model linearized after 10 steps tracks the non-linear performance in the catapult phase up to $\eta \approx 4/\lambda_0$.

Models evolve as linear models when the NTK is constant. In Figure S18b we give evidence that as networks become wider, the change in the kernel decreases.

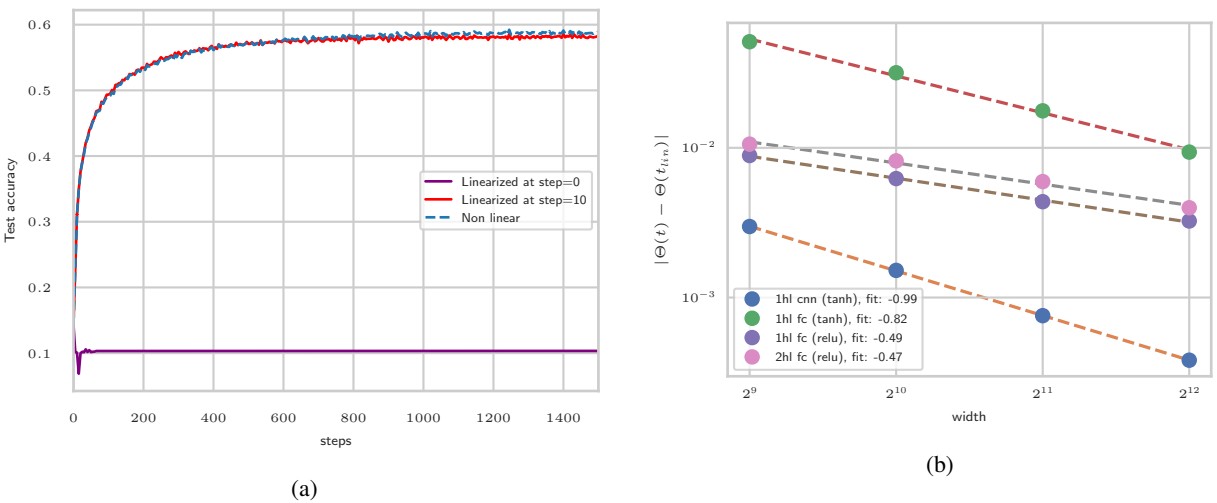

(a)

(b)

Figure S18: Evidence for a return of linear dynamics after $t_{\text{lin}}$. (a,b) Show the same model as in figure S4 with the addition of linearized models at step 0 and 10. We observe that the linearized model after 10 steps tracks the non-linear performance in the 'catapult' phase up to $\eta \sim \frac{4}{\lambda_0}$ (c) The change in the NTK between $t_{\text{lin}} = 50$ steps and $t = 1000$ steps decreases as the width increases. Here we consider 2-class MNIST with 100 samples per class.

