# OpenReview forum: "The large learning rate phase of deep learning"
_ICLR.cc/2021/Conference — Reject_

### Official Review · AnonReviewer2 · 2020-10-29
**The Large Learning Rate Phase of Deep Learning**

**Rating:** 3
**Confidence:** 4

**Review:**

This paper analyzes the effect of choosing a large step-size on the generalization of deep networks. They suggest that starting with a large learning rate, the loss initially increases before converging to a flatter minimum with improved generalization (catapult effect). When the learning rate is above a certain threshold, the authors observed this catapult effect along with a decrease in the curvature of the landscape. These observations were empirically demonstrated through various experiments and analytically studied for a two-layer linear network.

Pros:
The paper tackles an interesting problem in a vibrant field of research.

Several experiments that demonstrate the authors claims were presented in the paper. These results can be further used to propose algorithms designed to converge to flatter solutions.

The related material is referenced and well-discussed in the paper.

Cons:
This paper lacks theoretical evidence for supporting the raised claims. The model studied in Theorem 1 is very simple. Moreover, the effect of choosing the step-size and batch-size on the sharpness of the computed solution has been observed in the literature. For instance, Keskar et al. (2017) justified the improved generalization achieved when using small batch sizes by the ability of such methods to converge to flat minima. The choice of the batch-size was formally related to the choice of the step-size in the work of Patel (2017) which provides a learning-rate lower bound threshold for the divergence of batch SGD. This threshold is also a function of the curvature. The higher the curvature around a critical point, the lower the threshold required for divergence. Hence choosing a large step-size tend to escape sharp minima and potentially converge to flatter minimizers. The last paper was not cited in the submitted manuscript.

The paper does not provide any intuition on how to compute the thresholds $\mu_{\mbox{crit}}$ and $\mu_{\max}$, and how is this related to the choice of non-linearity, structure of the network, …

Comments:
1.	For the gradient descent update rule in equation (1): why do we have $f_t$?

Minor Comments:
1.	What is meant by ``compute budget’’?
2.	In the definition of $\Theta$ in Page 4: I think it should by $\sum_{\alpha = 1}^m$ instead of $\sum_{\mu=1}^p$.
3.	In Figure S1, the images are small and not very clear.
4.	Expression (S12) is missing a term.
5.	S(16), what is $u_{ia}$.
6.	Figure S10, the colors in the label do not match the plots.

---

> ### Author Response · Authors · 2020-11-17
> **Author response**
>
> We thank the reviewer for their detailed comments.
>
> > Cons: This paper lacks theoretical evidence for supporting the raised claims. The model studied in Theorem 1 is very simple. Moreover, the effect of choosing the step-size and batch-size on the sharpness of the computed solution has been observed in the literature. For instance, Keskar et al. (2017) justified the improved generalization achieved when using small batch sizes by the ability of such methods to converge to flat minima. The choice of the batch-size was formally related to the choice of the step-size in the work of Patel (2017) which provides a learning-rate lower bound threshold for the divergence of batch SGD. This threshold is also a function of the curvature. The higher the curvature around a critical point, the lower the threshold required for divergence. Hence choosing a large step-size tend to escape sharp minima and potentially converge to flatter minimizers. The last paper was not cited in the submitted manuscript.
>
> While the relation between step size and curvature has been studied before, to our knowledge our paper is the first to point out a sharp transition between the small and large learning regimes, which happens independently of SGD noise, as well as to explain the mechanism underlying the curvatures reached in these two regimes. We thank the reviewer for bringing Patel (2017) to our attention; we would like to note that the nonconvexity in our problem is particular to the structure of (wide) neural networks, and does not correspond to a landscape with multiple, separated quadratic minima, like the case studied in that work. Instead, what we find is that the landscape has a set of connected global minima of varying curvatures. Therefore, our results do not follow from the papers mentioned in the comment.
>
> > The paper does not provide any intuition on how to compute the thresholds $\mu_{\rm crit}$ and $\mu_{\rm max}$, and how is this related to the choice of non-linearity, structure of the network, …
>
> We believe the reviewer is referring to the thresholds $\eta_{\rm crit}$ and $\eta_{\rm max}$. $\eta_{\rm crit}$ is a simple function of the curvature at initialization (defined precisely in the paper). Understanding how the curvature relates to choices in network architecture and training data is a hard question and the subject of ongoing research in the community; however, operationally we can measure this curvature in the standard way by computing the top eigenvalue of a relatively small matrix. As for $\eta_{\rm max}$, based on our empirical results we suggest a heuristic for ReLU networks (given by $12/\lambda$), again in terms of the curvature $\lambda$ at initialization.
>
> > For the gradient descent update rule in equation (1): why do we have $f_t$
>
> Equation (1) lists the gradient descent update equations. The factor $f_t$ appears due to the loss function being $L=f_t^2/2$, and the fact that we have a factor of $dL/df_t = f_t$ when computing the gradient using the chain rule.
>
> > What is meant by “compute budget”?
>
> By this we mean the number of epochs used to train the model.
>
> > In the definition of Theta in Page 4: I think it should by (sic) $\sum_\alpha$ instead of $\sum_\mu$
>
> The NTK is defined by summing over the parameter index $\mu$, not the sample index $\alpha$.
>
> > Expression (S12) is missing a term.
>
> Can you please clarify which term is missing?
>
>
> > (S16), what is $u_{ia}$.
>
> It is the first layer weight matrix in the full model setting of Appendix B.3, where $u \in R^{n \times d}$ and the inputs $x \in R^{d}$.
>
> >Figure S10, the colors in the label do not match the plots.
>
> In this figure, different colors refer to different configurations, while dashed vs. solid patterns refer to what is being measured.

---

### Official Review · AnonReviewer3 · 2020-10-29
**Limits of large learning rate training at fixed learning rate SGD**

**Rating:** 4
**Confidence:** 4

**Review:**

The paper is a detailed account of how large of a learning rate a given mode can take when it is trained by constant step size SGD. Many papers investigated the effect of learning rate (and batch size and width etc…) to the final accuracy. In general, the findings indicate increased accuracy up to a certain threshold (where the algorithm doesn’t converge any more). Such findings are abundant in the literature. However, the present paper investigates the path that SGD takes to get to find those ‘good’ performing points in the weight space. Two phases emerge: monotonic decay in loss vs catapult regime and the latter appears to perform the best.
The paper has interesting findings but also some shortcomings, in more detail:
- Similar curve as in Figure 14 of https://arxiv.org/abs/1905.03776 as Fig 1 of the present paper, in that sense the findings of large lr better are certainly not novel, but the authors are well aware of this
- The paper defines the curvature as the max eigenvalue of the Fisher information matrix. The large learning rate values are determined according to that value. Fig 2 c and f show the cross over of the regimes. What would make the cross over sharper? Increasing width? Larger batch size?
- The large increase in loss value is huge! It is surprising that training doesn’t just diverge from there. What controls the growth and the subsequent decay? How consistent is it? Would the result hold in all random seeds or what fraction would still converge?
- What’s the max tr loss for the wide res net? (Why not consistent plots for fully connected and resnets in fig2?) (Also, for clarity, Fig 2 label of B would be Max tr loss)
- Fig 3 would be clearer if the test error axis of (a,b) would be aligned, same for (c,d). It appears as if decay actually makes the search for the correct large lr rate redundant. (Perhaps at the expense of searching for the right decay rate). In general, what can we do to maximize the range of hyperparameters that works for the model (to minimize search)? The paper proposes a heuristic based on the initial local curvature yet the findings for practice seem somewhat preliminary.
- The comparison between lazy training and how large learning rate deviates from lazy training is a very interesting and active area of research. I think the paper would be more complete if the experiments would verify it in a more robust way. Measuring the deviation of the activations or the movement of the kernel would be helpful. Here is one of the references with clear observables to measure in the experimental framework https://arxiv.org/abs/1906.08034

The characteristic difference in the early time dynamics is claimed to be related to the performance gains. There is a growing body of evidence that points to the role of the initial dynamics of SGD when it comes to identifying the final performance. The paper is further evidence in that direction. Therefore I think it’ll be useful for the ICLR community. Overall, the paper has very interesting bits and pieces but it fails to come together as a coherent whole to provide a consistent story.

---

> ### Author Response · Authors · 2020-11-17
> **Author response**
>
> We thank the reviewer for their detailed review.
>
> > The paper defines the curvature as the max eigenvalue of the Fisher information matrix. The large learning rate values are determined according to that value. Fig 2 c and f show the cross over of the regimes. What would make the cross over sharper? Increasing width? Larger batch size?
>
> According to our theoretical analysis, the cross-over can be made sharper by increasing the width.
>
> > The large increase in loss value is huge! It is surprising that training doesn’t just diverge from there. What controls the growth and the subsequent decay? How consistent is it? Would the result hold in all random seeds or what fraction would still converge?
>
> Indeed, the increase in loss value in the catapult phase is huge. In our experiments this result is consistent across runs, as long as the learning rate is not chosen to be too large (i.e. is not too close to the maximum learning rate). Our theoretical analysis indicates that the loss reaches a value proportional to the width, and also clearly explains why training does not diverge: while the loss grows, the curvature shrinks, and once the curvature is small enough the loss starts to decrease. In the theoretical model this result holds deterministically. Empirically we have also found this to consistently hold across random seeds, unless one is close to the maximum learning rate.
>
> > What’s the max tr loss for the wide res net? (Why not consistent plots for fully connected and resnets in fig2?) (Also, for clarity, Fig 2 label of B would be Max tr loss)
>
> In Figure 2 we wanted to show several different empirical effects related to loss and curvature early on in training, which is why we plot slightly different data for the two networks. However, in Figure 2.d the catapult effect is clearly visible in the Wide ResNet, and Figure 2.f shows that the curvature undergoes the phase transition at the predicted critical learning rate.
>
> Thank you for pointing out the typo in the label of Figure 2.b; we will correct it.
>
> > Fig 3 would be clearer if the test error axis of (a,b) would be aligned, same for (c,d). It appears as if decay actually makes the search for the correct large lr rate redundant. (Perhaps at the expense of searching for the right decay rate). In general, what can we do to maximize the range of hyperparameters that works for the model (to minimize search)? The paper proposes a heuristic based on the initial local curvature yet the findings for practice seem somewhat preliminary.
>
> The errors-axis of figure 3 is chosen so that the performance for the different learning rates is displayed in the plot. As the reviewer mentions, performance of small learning rates can be competitive with large learning rates if they are evolved for a time proportional to 1/eta (what we call physical time), which tends to be a very long time for practical learning rates. However, in the Wide ResNet model, the best performance happens for learning rates in the catapult phase even when training for this amount of time.
>
> > The comparison between lazy training and how large learning rate deviates from lazy training is a very interesting and active area of research. I think the paper would be more complete if the experiments would verify it in a more robust way. Measuring the deviation of the activations or the movement of the kernel would be helpful. Here is one of the references with clear observables to measure in the experimental framework https://arxiv.org/abs/1906.08034
>
> We thank the reviewer for this suggestion and the reference. We would like to point out that in the paper we focus on presenting results for the maximum eigenvalue of the Neural Tangent Kernel (equivalently of the Fisher Information Matrix), for consistency with the theoretical model. The fact that this eigenvalue changes considerably in the large learning rate phase implies that the NTK changes considerably, and therefore shows that lazy training does not occur in this phase. However, we have also empirically measured the kernel movement, and it displays the same sharp behavior as a function of learning rate when networks are made wider.

---

### Official Review · AnonReviewer4 · 2020-10-31
**Valuable insights on a highly relevant problem, contributions are somewhat limited**

**Rating:** 5
**Confidence:** 4

**Review:**

The paper studies the effect of the learning rate's magnitude when training neural networks. I believe this to be an extremely relevant problem since large learning rates are widely adopted in practice due to the their positive impact on the model's generalization, even though we don't understand the reason behind this.

The main contributions come from the analysis of a 2-layer linear network trained on a single data point, which shows the existence of three regimes that are governed by the learning rate's magnitude and the local curvature at initialization. Unlike prior work that also identifies different regimes governed by the learning rate, this submission does not point at noise as the main responsible for the regimes.

Presented experiments are interesting and clearly show the different regimes and how they are connected to the learning rate, but do not seem to provide significant new insights compared to previous work. I believe the plots showing how the curvature behaves as the learning rate varies are novel to the best of my knowledge, but are not surprising given the previous discussions on how the learning rate affects the solution's sharpness.

While the theoretical result does provide a precise characterization of the phenomena, it is restricted to a very constrained setting (2-layer linear net trained on one data point), making it hard to evaluate how the result translates to more complex settings (although the authors do a reasonable job at exploring this via experimental analysis).

I also find the discussion on the presented theory to be somewhat lacking. For example, while the paper seems to downplay the importance of noise in characterizing the three regimes (compared to prior work), the curvature inevitably changes if noise is added to the sample gradients, e.g. adding isotropic noise will decrease \eta_crit and shift the phase transitions, which is also expected to occur if as we decrease the batch size when doing mini-batch SGD. The fact that different batch sizes have been adopted across experiments (even for the same dataset/model) makes the experimental results harder to evaluate when compared to the developed theory. Except for a footnote on page 2, there is no discussion on how the initialization scale affects the phase transitions, which would be useful to better understand how the theory relates to prior work that focuses on the init scale instead of noise or width.

Overall I believe the problem to be very relevant and the approach to be promising, but the theoretical results are overly limited and the contributions are below what I'd expected to suggest acceptance. Since extending the analysis to non-linear networks would seem to be a considerable technical leap, the authors could instead extend it to a less limited training setting where gradient variance is non-zero, in which case \eta_crit would have a dependence on the batch size assuming mini-batch SGD updates.

----

I have read the response and am keeping my score. I agree that the simplicity of the results/model is valuable, but additional theoretical results (even extensions to Theorem 1, with more involved but stronger claims) would greatly improve the paper and make its contributions closer to what is expected of a ICLR submission. Extending the result to consider noise should be straightforward and yield a reasonably simple claim, which can be further verified empirically by adding synthetic noise to the gradients, adding label noise, and/or adopting extremely small batch sizes. As it stands now, the submission is still lacking in terms of contributions.

---

> ### Author Response · Authors · 2020-11-17
> **Author response**
>
> We thank the reviewer for their thoughtful and detailed comments.
>
> As the reviewer points out, our theoretical results are shown in the simple setting of a 2-layer linear network. However, we show empirically that the detailed behavior of this model matches that of deep networks with common fully-connected and convolutional architectures.
>
> In light of this, we argue that the simplicity of the model is an advantage rather than a drawback. Our model is able to explain quantitative phenomena -- the sharp transition in loss and curvature dynamics that occurs at the critical learning rate. If we share the goal of developing a better understanding of deep learning phenomena, then the best model is the simplest model that can explain the phenomena. Indeed, thanks to the simplicity of the model, we were able to develop a detailed understanding of the underlying mechanism that gives rise to these effects. That being said, we do extend the model to the case of multiple data points in supplemental section B.3.
>
> While the sharp dependence of curvature on learning rate is one contribution of our work, there are others we discuss. (1) We show the sharp drop in curvature, which occurs in the catapult regime, always occurs simultaneously with a large spike in the loss. (2) We discuss how the (i) maximum loss and (ii) step at which the catapult effect occurs grow with width (respectively, linearly and logarithmically). (3) We give a theoretical prediction for the maximum learning rate beyond which gradient descent diverges and show empirically that this holds for networks with non-ReLU nonlinearities. (4) Network width is an important ingredient which controls the sharpness of the phenomenon we study. Our work gives evidence for a regime of nonlinear (non-NTK) dynamics that exists in networks that are arbitrarily wide.
>
> Regarding the effect of SGD noise, we agree that the noise shifts the positions of the phase transitions. However, in practice we find that the shift is mild, as can be seen in the empirical results: even though we train most networks using SGD rather than full-batch gradient descent, we observe the phase transitions close to the critical learning rate predicted by the full-batch analysis. We therefore chose not to explore the shift due to noise in great detail in this paper.

---

### Decision · Program_Chairs · 2021-01-07
**Final Decision**

**Decision:**

Reject

**Comment:**

I agree with the reviewers that said that this paper has valuable insights. However, all reviewers ultimately recommended rejection. I think the main reason was that the reviewers did not feel these insights don't accumulate together to a message that would justify a paper. I hope the authors can address these concerns and resubmit. There were additional concerns, like the fact a very simplistic toy model is being used, but I agree with the authors that it makes sense to first explore such phenomena in the simplest model that produces them.